# FinSight: Towards Real-World Financial Deep Research

## Abstract

Generating professional financial reports is a labor-intensive and intellectually demanding process that current AI systems struggle to fully automate. To address this challenge, we introduce FinSight (**Fin**ancial In**Sight**), a novel multi-agent framework for producing high-quality, multimodal financial reports. The foundation of FinSight is the Code Agent with Variable Memory (CAVM) architecture, which unifies external data, designed tools, and agents into a programmable variable space, enabling flexible data collection, analysis and report generation through executable code. To ensure professional-grade visualization, we propose an Iterative Vision-Enhanced Mechanism that progressively refines raw visual outputs into polished financial charts. Furthermore, a Two-Stage Writing Framework expands concise Chain-of-Analysis segments into coherent, citation-aware, and multimodal reports, ensuring both analytical depth and structural consistency. Experiments on various company and industry-level tasks demonstrate that FinSight significantly outperforms all baselines, including leading deep research systems in terms of factual accuracy, analytical depth, and presentation quality, demonstrating a clear path toward generating reports that approach human-expert quality. Our code is available at https://anonymous.4open.science/r/FinSight-6739/.

## 1 Introduction

Investment decisions worth billions of dollars hinge on the quality and timeliness of financial research reports (Tian et al., 2025). These reports translate raw market data into strategic insights, serving as analytical support for asset managers, equity researchers, and institutional investors. However, producing such reports remains a challenging task due to the overwhelming volume of financial data and the demand for rapid, high-quality analysis (Ren et al., 2021; Jimeno-Yepes et al., 2024). Recent advances in artificial intelligence, particularly in reasoning models (OpenAI, 2024; DeepSeek-AI et al., 2025; Bai et al., 2025), deep search and research applications (OpenAI, 2025a; Gemini, 2025; Grok, 2025; Camara, 2025), present great potential in solving these labor-intensive collecting and analyzing tasks. Despite these technical advances, significant challenges persist in automating the generation of full financial research reports that meet the high standards for data accuracy, analytical depth, and multimodal content integration (Yang et al., 2025).

Existing methods face several limitations that hinder their practical adoption: **(1) Lack of Financial Domain Knowledge:** Most current systems, whether closed-source (OpenAI, 2025a; Grok, 2025; Gemini, 2025) or open-source (Hu et al., 2025; Li et al., 2025b), are designed for general search scenarios, ignoring the integration of real-time heterogeneous financial data (both unstructured articles, news, and structured data). **(2) Limited Multimodal Support and Visualization:** Almost all current methods can only produce plain-text reports, lacking diverse visualizations (*e.g.*, figures, charts and tables) that are critical in conveying information (Yang et al., 2025). **(3) Insufficient Analytical Depth:** Current methods often rely on rigid, predefined workflows for single-pass data collection (Trivedi et al., 2023; Li et al., 2025a; Jin et al., 2025) and report generation (Chen et al., 2024), preventing them from dynamically adjusting research strategies based on intermediate findings, ultimately limiting the analytical depth and insight of the final report.

To address these challenges, we introduce **FinSight**, a novel multi-agent system that simulates the cognitive processes and analytical workflows of expert financial researchers. FinSight operates three

necessary stages: (1) Data Collection, which gathers up-to-date heterogeneous data and organizes it into a structured multimodal memory. (2) Data Analysis, where an interactive environment enables multi-round interactions with data, tools, and agents to derive a concise Chain-of-Analysis sequence. (3) Report Generation, which follows a draft outline to transform the data and Chain-of-Analysis into a formatted financial report with chart and data references, finally rendered in a professional style.

To realize FinSight, we reconstruct the deep research workflow and propose a novel agent architecture, **Code Agent with Variable Memory (CAVM)**, where all data, tools, and agents are unified into a programmable variable space accessible and manipulable through executable code. This architecture leverages the code capabilities of language models (Wang et al., 2024a; Jiang et al., 2024; Tang et al., 2024), and enables flexible, scalable task handling from bottom-up data operations to high-level workflow orchestration.

To address the critical challenges of multimodal generation and analytical depth, we introduce two specialized mechanisms. To overcome the shortcomings of automated visualization, we propose an Iterative Vision-Enhanced Mechanism, where a vision-language model provides critical feedback to iteratively refine code-generated charts until they meet professional standards. For the challenge of generating coherent, long-form reports, we employ a Two-Stage Writing Framework. This framework first distills insights into concise Chain-of-Analysis segments, which then serve as a structured foundation for the Report Generation Agent to compose a full, context-aware report with tightly integrated visualizations and citations. Our extensive evaluations demonstrate that this synergistic approach enables FinSight to significantly outperform existing methods, delivering reports with superior accuracy, depth, and multimodal coherence.

To comprehensively evaluate our method, we construct a high-quality benchmark featuring research tasks at both company and industry levels, spanning multiple markets and diverse sectors. Utilizing this benchmark, we conduct both LLM-as-a-Judge and human evaluations, with a specific focus on fine-grained assessments of factuality and citation accuracy. Experiments demonstrate that our method significantly surpasses various deep research systems across three key dimensions: **Factual Accuracy**, **Analytical Depth**, and **Presentation Quality**, validating that FinSight can generate rich, insightful, and multimodal financial research reports that approach the quality of human experts.

Our core contributions are as follows:

1. We propose a novel multi-agent framework for **Multimodal Deep Research**. To the best of our knowledge, this work presents the first exploration of **Multimodal Deep Research** capable of generating long-form reports with **interleaved text and charts**. While benchmarked in the financial domain, our work establishes a generalizable paradigm for future deep research systems, extending the boundary from text-only search to comprehensive multimodal content generation.

2. We design the **Code Agent with Variable Memory (CAVM)** architecture to instantiate this framework. By unifying data, tools, and agents into a programmable variable space, CAVM enables the flexible and scalable workflow orchestration required for complex tasks.

3. We propose an **Iterative Vision-Enhanced Mechanism** for professional chart generation that integrates the code-generation capabilities of large language models with the visual understanding of vision–language models to iteratively refine basic charts into professional-quality visualizations.

4. We introduce a **Two-stage Writing Framework with Generative Retrieval** that progresses from short and concise Chain-of-Analysis segments to long and comprehensive financial reports, seamlessly integrating textual analysis with visual elements to meet the need for real-world financial multimodal deep research.

## 2 METHOD

### 2.1 PROBLEM FORMULATION

We formalize the task of *Professional Financial Report Generation* as an open-ended, multimodal generation process. Unlike general web summaries, this task targets investment-grade research standards, requiring high analytical depth, rigorous data collection, and multimodal synthesis (*e.g.*,

Figure 1: Overview of the FinSight Framework.

integrating professional charts with insights). Given a research query $q$ (*e.g.*, *"Analyze the competitive landscape of the EV battery industry"*), the system aims to generate a structured report $R$. To address the complexity of long-form writing, we model $R$ not as a disordered bag of words, but as a hierarchical ordered sequence:

$$R = \{S_1, S_2, \ldots, S_N\},$$

where $N$ is the number of sections derived from a dynamic outline $\mathcal{O}$. Each section $S_i$ is further defined as an ordered sequence of multimodal elements $S_i = (e_{i,1}, e_{i,2}, \ldots, e_{i,m})$, where each element $e_{i,j} \in \{T, V, C\}$ represents text segments, visualization figures, or citations, respectively.

## 2.2 THE FRAMEWORK OF FINSIGHT

FinSight is a multi-agent system designed to simulate the workflow of a professional financial analyst. The system realizes three core processes: multi-source data collection, multi-turn data analysis and progressive report writing, implemented through the CAVM architecture described in Section 2.3. The key design of this framework will be detailed in the following sections.

**Data Collection** To address the limitations of general web search systems in financial domains, we design two specialized agents for comprehensive data gathering: (1) **Deep Search Agent:** Conducts iterative, multi-round investigations using search engines and virtual browsers to gather comprehensive information with source verification. (2) **Multi-Source Data Collection Agent:** Collects heterogeneous data from financial databases, APIs, and web sources, leveraging different tools to access diverse information types. It can invoke the deep search agent for specific information requirements. Instead of treating data collection as an isolated preliminary step, FinSight allows the analysis and writing stages to dynamically invoke further data collection, ensuring broader and more relevant knowledge coverage.

**Data Analysis** Built on CAVM, the **Data Analysis Agent** executes analytical tasks via multi-turn code actions, dynamically deciding when to process data, invoke data collection workflows, or terminate with a concise Chain-of-Analysis (CoA) output (Section 2.5). It integrates the Iterative Vision-Enhanced Mechanism (Section 2.4) for professional chart generation.

**Report Generation** The **Report Generation Agent** handles drafting, optimization, and post-processing using the Two-Stage Writing Framework (Section 2.5). The process includes: (1) *Drafting:* retrieving relevant CoA segments and structured data according to predefined outlines; (2) *Self-reflective Optimization:* iteratively refining text for factual accuracy and consistency; and

(3) *Post-processing:* parsing identifiers, loading visualizations, formatting citations, and rendering into a publication-ready format.

## 2.3 CODE AGENT WITH VARIABLE MEMORY (CAVM)

**Motivation: From Reading Context to Manipulating Variables** Traditional agents typically rely on unstructured text or vector embeddings as memory. While sufficient for general tasks, this paradigm struggles in professional financial scenarios which require precise calculations and handling of massive heterogeneous data. To address this, we propose **Code Agent with Variable Memory (CAVM)**, a novel architecture that redefines agent memory as a *Programmable State Representation*. The core philosophy is to shift the agent's interaction mode from *reading context* to *manipulating variables*. This design empowers agents

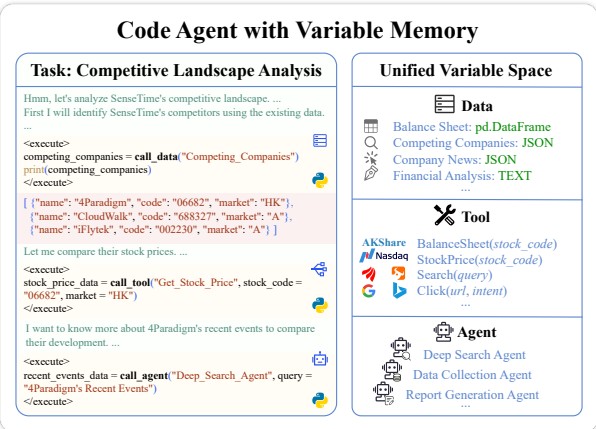

Figure 2: The design philosophy of CAVM architecture.

to maintain data as executable objects (e.g., DataFrames) rather than static text, enabling rigorous mathematical operations and significantly reducing hallucinations in numerical reasoning.

**Unified Variable Space** We abstract the multi-agent collaboration environment into a unified variable space $\mathcal{V}$, encompassing three distinct types as shown in Figure 2: (1) **Data ($\mathcal{V}_{data}$):** Stores both structured (e.g., 'pandas.DataFrame' for financial tables) and unstructured data as executable Python objects; (2) **Tools ($\mathcal{V}_{tool}$):** Functional interfaces for external interaction; (3) **Agents ($\mathcal{V}_{agent}$):** encapsulated agent instances that can be invoked recursively.

$$\mathcal{V} = \mathcal{V}_{data} \cup \mathcal{V}_{tool} \cup \mathcal{V}_{agent}.$$

This unified scope allows heterogeneous elements to be accessed via a standard code interface. For instance, an agent can perform statistical analysis on $\mathcal{V}_{data}$ or invoke another expert agent from $\mathcal{V}_{agent}$ within the same code block, supporting hierarchical reasoning that static context windows cannot achieve.

**Foundation Agent with Code Action** Built upon this variable space, the agent operates in an iterative loop of reasoning and code execution. Unlike purely generative agents, our agent actively decides which variables to retrieve or modify via code, ensuring **contextual conciseness**. Formally, at step $t$, the agent generates a reasoning trace $\mathcal{R}_t$ and a code action $\mathcal{C}_t$:

$$P_\theta(\mathcal{R}_t, \mathcal{C}_t \mid q, \mathcal{V}_{t-1}, \mathcal{H}_{t-1}) = \underbrace{P_\theta(\mathcal{R}_t \mid \Phi(\mathcal{V}_{t-1}), \cdot)}_{\text{Reasoning}} \cdot \underbrace{P_\theta(C_t \mid \mathcal{R}_t, \Phi(\mathcal{V}_{t-1}), \cdot)}_{\text{Code Action}},$$

where $\Phi$ is a formatting function that summarizes the metadata of variables in $\mathcal{V}_{t-1}$. The code $\mathcal{C}_t$ is then executed by a Python interpreter to update the variable space:

$$\mathcal{V}_t, \text{output}_t = \text{Execute}(\mathcal{C}_t, \mathcal{V}_{t-1}), \tag{1}$$

$$\mathcal{H}_t = \mathcal{H}_{t-1} \oplus \text{output}_t. \tag{2}$$

This mechanism allows the agent to maintain a "working memory" of precise data states throughout long-horizon tasks.

## 2.4 ITERATIVE VISION-ENHANCED MECHANISM FOR VISUALIZATION

**Motivation** Generating high-quality visualizations is a persistent challenge in automated report generation, particularly in data-intensive domains like finance that require nuanced analysis and

presentation. Existing methods often rely on single-pass code execution or employ Vision-Language Models (VLMs) without incorporating visual feedback, which frequently leads to suboptimal outcomes. Drawing inspiration from Chain-of-Thought (Wei et al., 2022) and Actor-Critic (Schulman et al., 2017), we propose a framework where an agent learns to progressively improve visualizations. This is achieved by iteratively plotting a chart and refining it based on critical feedback, ensuring both stable generation and continuous quality enhancement.

**Iterative Vision-Enhanced Mechanism** Specifically, the final output of the Data Analysis Agent includes the target chart specifications along with the corresponding descriptions and data. For each chart, the agent generates an initial visualization through executable plotting code, which is then evaluated by a VLM to give potential issues of visual cues (e.g. missing labels, inappropriate color schemes). These feedbacks are sent to the system, directing the iterative code generation until the output reaches professional quality.

$$P(\mathcal{C}_{vis} \mid \mathcal{V}) = \prod_{t=1}^{M} P_\theta(\mathcal{C}_t^{vis} \mid \mathcal{C}_{t-1}^{vis}, \mathcal{F}_{t-1}, \mathcal{V}), \quad \mathcal{F}_{t-1} = \text{VLM}(\text{Execute}(\mathcal{C}_{t-1}^{vis})),$$

where $M$ is the maximum number of iterations. The iteration continues until convergence or a predefined quality threshold is satisfied.

## 2.5 TWO-STAGE WRITING WITH GENERATIVE RETRIEVAL

**Motivation** A complete report encompasses analyses from multiple perspectives, which can be regarded as an integration of several Chains-of-Analysis. To generate long-form financial research reports with both textual depth and multimodal coherence, we design a **two-stage writing framework** augmented with generative retrieval. It decomposes the report writing process into (1) Chain-of-Analysis Generation and (2) Structured Writing with Generative Retrieval.

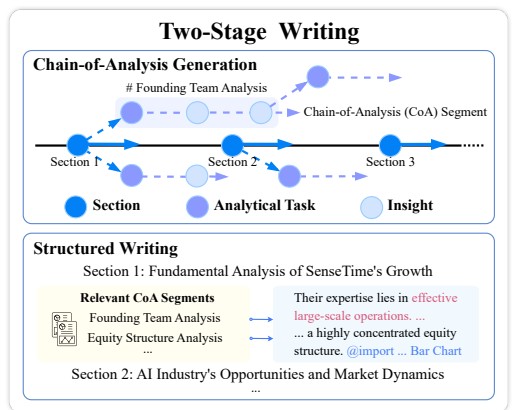

Figure 3: Chain-of-Analysis Illustration.

**Stage 1: Chain-of-Analysis Generation** Given the research question $q$, the Data Analysis Agent first generates a set of analytical perspectives $\mathcal{P} = \{p_1, p_2, ..., p_K\}$. The agent then performs parallel data analysis for each $p_i$, producing corresponding Chain-of-Analysis (CoA) that capture insights from distinct viewpoints.

Each CoA is generated based on the interaction history $\mathcal{H}_i$, accumulated during the data analysis process. To ensure coherence between textual content and referenced elements (e.g. figure, reference), this process is augmented with a generative retrieval mechanism that jointly produces textual contents along with element identifiers. These identifiers specify chart and reference attributes using natural language descriptions, enabling unified autoregressive generation. The process can be formalized as:

$$P(\mathcal{A} \mid q, \mathcal{V}) = P(\mathcal{P} \mid q, \mathcal{V}) \cdot \prod_{i=1}^{|\mathcal{P}|} P(a_i \mid p_i, \mathcal{V}).$$

**Stage 2: Structured Writing** Building on CoAs, a Report Generation Agent first constructs a report outline $\mathcal{O} = \{o_1, o_2, ..., o_n\}$, and then writes each section sequentially. For each section $s_i$, the agent dynamically retrieves the most relevant data and CoA segments from the unified variable memory $\mathcal{V}$, formalized as:

$$P(R \mid \mathcal{A}, \mathcal{V}, q) = P(\mathcal{O} \mid \mathcal{A}, q) \cdot \prod_{i=1}^{n} P(A_{\text{selected}}^{(i)}, \mathcal{V}_{\text{selected}}^{(\langle\rangle)} \mid \mathcal{A}, \mathcal{V}, \cdot) \cdot P(s_i \mid s_{<i}, A_{\text{selected}}^{(i)}, \mathcal{V}_{\text{selected}}^{(\langle\rangle)}, \cdot).$$

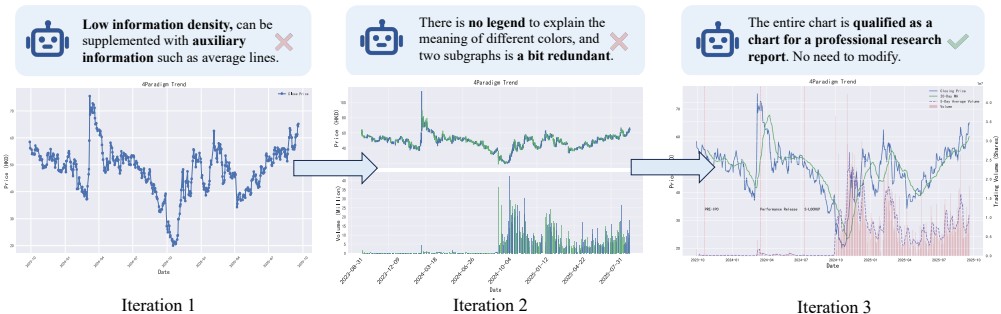

Iteration 1         Iteration 2         Iteration 3

Figure 4: An example of our Iterative Vision-Enhanced Mechanism of Visualization. The chart is generated by matplotlib and seaborn package in Python.

To prevent hallucination of non-existent references and figures, agent is instructed to follow the identifiers established in $\mathcal{A}$. To ensure reference accuracy, the agent strictly follows the identifiers established during the stage 1.

## 3 EXPERIMENTS

### 3.1 DATASET AND EVALUATION METRICS

Financial research report generation remains an under-explored problem lacking appropriate evaluation benchmarks and metrics. To address this gap, we construct a high-quality benchmark specifically designed for financial research report generation, comprising a dataset of research targets with corresponding professional institutional reports and a comprehensive set of automated evaluation metrics. Details can be found in Appendix C.6.

**Dataset.** Our dataset encompasses research targets at both company and industry levels. For company-level analysis, we curated a diverse list of companies from authoritative financial platforms, covering different markets, industry sectors, and market capitalizations. For industry-level analysis, we selected high-attention industries from these platforms as research targets. For all targets, we collected in-depth analysis reports authored by professional brokerage institutions as golden reference reports to facilitate evaluation of data accuracy and analytical quality.

To ensure the quality of golden reference reports, we applied stringent filtering criteria, selecting only reports exceeding 20 pages in length and containing more than 20 charts and visualizations. Following established practices in report generation research (Wang et al., 2024b; 2025; Li et al., 2025b), and considering the substantial time and computational costs associated with report generation and evaluation, we collected 20 samples: 10 company-level and 10 industry-level targets.

**Evaluation Metrics.** We design 9 automated evaluation metrics across three critical dimensions, each ranging from 0 to 10 points. Detailed description of each metric can be found in Appendix C.6.

**(1) Factual Accuracy:** Measures the reliability and correctness of generated content through Core Conclusion Consistency (alignment with reference conclusions), Textual Faithfulness (proper citation support), and Text-Image Coherence (consistency between textual and visual elements).

**(2) Information Effectiveness:** Evaluates the analytical value delivered to investors via Information Richness (distinct information points), Coverage (proportion of key reference information captured), and Analytical Insight (critical analysis and forward-looking recommendations).

**(3) Presentation Quality:** Assesses professional standards through Structural Logic (organizational coherence), Language Professionalism (adherence to financial terminology), and Chart Expressiveness (effective visualization utilization and aesthetic quality).

Table 1: Overall evaluation results on financial report generation benchmark (averaged over three runs). **Bold** denotes the highest score in each column, Underlined denotes the second highest.

| Model | Factual | | | Analytical | | | Presentation | | | Avg. |
|---|---|---|---|---|---|---|---|---|---|---|
| | Cons. | Faith. | T-I. | Rich. | Cover. | Ins. | Logic | Lang. | Vis. | |
| *LLM with Search Tools* | | | | | | | | | | |
| GPT-5 w/ Search | 5.95 | 6.35 | 4.77 | 5.43 | 4.52 | 5.09 | 6.53 | 5.87 | 3.90 | 5.38 |
| Claude-4.1-Sonnet w/ Search | 5.78 | 5.92 | 3.55 | 5.58 | 5.25 | 5.01 | 6.34 | 6.07 | 2.59 | 5.12 |
| DeepSeek-R1 w/ Search | 6.26 | 5.92 | 4.08 | 6.68 | 6.33 | 6.62 | 7.03 | 6.79 | 3.35 | 5.90 |
| *Deep Research Agent* | | | | | | | | | | |
| Grok Deep Search | 4.71 | 5.72 | 4.21 | 4.90 | 4.03 | 4.35 | 5.87 | 5.61 | 3.76 | 4.79 |
| Perplexity Deep Research | 5.02 | 5.74 | 4.03 | 3.88 | 3.40 | 3.65 | 5.47 | 4.92 | 3.42 | 4.39 |
| Gemini-2.5-Pro Deep Research | 5.92 | 6.66 | 4.32 | 6.19 | 6.03 | 5.74 | 6.77 | 6.70 | 3.23 | 5.73 |
| OpenAI Deep Research | **6.87** | 6.78 | 4.58 | 6.79 | 6.83 | 7.33 | 7.56 | 7.58 | 3.66 | 6.44 |
| **FinSight (ours)** | 6.84 | **7.59** | **7.84** | **8.49** | **8.44** | **7.78** | **7.82** | **7.98** | **8.57** | **7.93** |

## 3.2 BASELINES

We compare FinSight against multiple categories of baselines:

**LLMs with Search Tools:** We evaluate leading large language models directly combined with search tools for report generation, including OpenAI GPT-5 (OpenAI, 2025b), DeepSeek-R1 (DeepSeek-AI et al., 2025), and Claude-4.1-Sonnet (Google, 2023).

**Deep Research Agents:** We compare against state-of-the-art commercial deep research products, including Gemini-2.5-Pro Deep Research (Gemini, 2025), Grok Deep Search (Grok, 2025), OpenAI Deep Research (OpenAI, 2025a), and Perplexity Deep Research[1]. Details of baseline implementations can be found in Appendix C.1.

## 3.3 IMPLEMENTATION DETAILS

Our backbone model uses DeepSeek-V3, and during the writing phase, we employ DeepSeek-R1 with reasoning capabilities. The maximum input length is set to 81,920, and the maximum output length is set to 16,384. For search, we use the Google Search API and retrieving the top 10 search results. For evaluation, we employ Gemini-2.5-Pro as our primary judge model. To ensure statistical robustness, we conduct three independent evaluation runs per sample and report mean scores with 95% confidence intervals. Additionally, we employ GPT-5 as an auxiliary evaluator to mitigate potential single-model bias and verify cross-model consistency.

To complement our automated metrics, we conducted a rigorous human evaluation involving **6 graduate students with backgrounds in finance**. In human evaluation part, we compared FinSight against the two strongest commercial baselines: Gemini-2.5-Pro Deep Research and OpenAI Deep Research. We report the inter-rater reliability using Krippendorff's $\alpha$ and the correlation between human and LLM judges using Pearson's $r$. Details can be found in Appendix C.

## 3.4 MAIN RESULTS

Table 1 presents the performance of FinSight against two categories of baselines on the financial research report generation task. Overall, FinSight achieves the highest overall score (8.09), significantly outperforming all baselines, including closed-source commercial agents like Gemini Deep Research (6.82) and OpenAI Deep Research (6.11). This result validates the effectiveness of our proposed multi-agent framework for crafting in-depth financial research reports. In terms of factuality, FinSight obtains the best scores in both the faithfulness of text citations and text-image consistency, demonstrating the efficacy of the identifier mechanism designed within our Chain-of-Analysis process.

---

[1] https://www.perplexity.ai/?model_id=deep_research

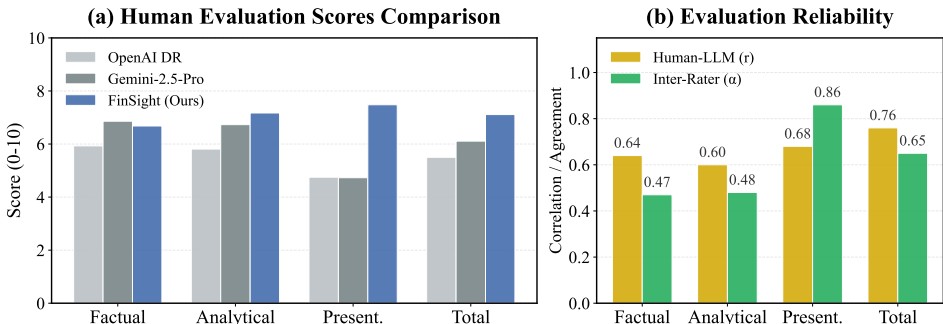

Figure 5: **Human Evaluation Results.** **(a)** Comparison of human scores (0-10 scale). FinSight (red bars) consistently achieves high scores, demonstrating a significant lead in Presentation and Analytical dimensions compared to commercial baselines. **(b)** Reliability analysis metrics. High Human-LLM correlation ($r$) validates our automated judges, and robust Inter-Rater reliability ($\alpha$) confirms the consensus among human experts.

A noteworthy observation is that the consistency score of our model (6.85) is slightly lower than that of Gemini Deep Research (7.10). Case studies reveal that our method prioritizes comprehensive data acquisition to deliver deeper insights. This approach leads it to uncover more data-driven findings, rather than generating simplified conclusions from conventional search-based methods.

The superiority of our method is further reflected in the analytical quality of the reports. FinSight scores the highest in information richness, coverage of key information from professional reports, and insightfulness. Regarding presentation quality, our system demonstrates a comprehensive lead in logic, language, and visualization. It particularly excels in visualization (9.00), far surpassing other methods and showcasing the advanced multimodal presentation capabilities of our system.

### 3.5 ABLATION STUDIES

We conduct ablation studies to evaluate the contribution of our key components, with results summarized in Table 2. Key findings are as follows: (1) Removing iterative VLM feedback for chart generation causes a significant decline in both Presentation Quality (from 8.0 to 7.5) and Analytical Quality (from 7.9 to 7.2). This is primarily because the writing process relies on analyzing the generated images, lower-quality visuals impede the ability to perform insightful analysis based on the charts. (2) Merging analysis and writing into a single process leads to a significant drop in analytical quality (from 7.9 to 5.9) and factual accuracy (from 7.0 to 6.4), demonstrating the

Table 2: Ablation studies of our key design.

| Method | Fact. | Ana. | Pres. |
|---|---|---|---|
| FinSight | 7.0 | 7.9 | 8.0 |
| w/o Iter.. | 6.9 | 7.2 | 7.5 |
| w/o 2-Stage. | 6.4 | 5.9 | 6.3 |
| w/o Dyn. | 5.9 | 5.7 | 6.4 |

effectiveness of our proposed two-stage, analyze-then-write strategy. (3) Eliminating dynamic search during the analysis and writing phases results in a significant performance drop across all dimensions, including Factual Accuracy (from 7.0 to 5.9) and Analytical Depth (from 7.9 to 5.7). This highlights the necessity of acquiring additional knowledge during these stages to ensure comprehensive and factually correct reports.

### 3.6 HUMAN EVALUATION

As illustrated in Figure 5, our key observations are: **(1) Superior Overall Performance:** Figure 5(a) presents the comparative results. FinSight achieves the highest total score (7.11), significantly outperforming Gemini-2.5-Pro (6.11) and OpenAI Deep Research (5.50). While Gemini-2.5-Pro shows competitive performance in the Factual dimension, FinSight establishes a substantial margin in *Analytical Depth* and *Presentation Quality*. Specifically, in the Presentation dimension, FinSight scores 7.48 compared to ¡5.0 for baselines, highlighting the impact of our multimodal chart generation capabilities. **(2) Validation of Automated Metrics:** Figure 5(b) details the reliability

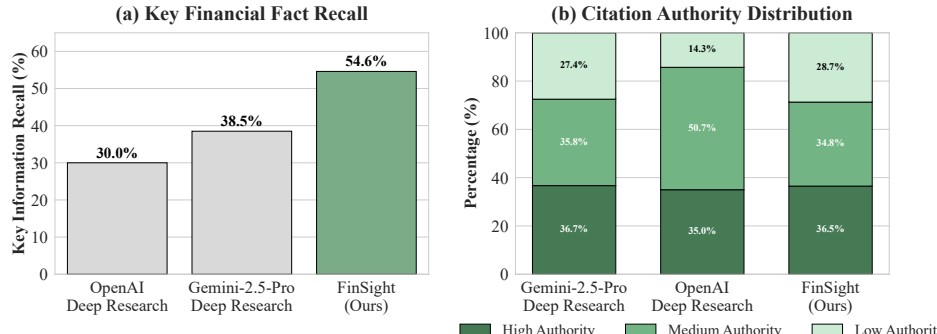

Figure 6: Factuality Evaluation. (a) **Key Financial Fact Recall**: The percentage of ground truth facts from golden reports covered by each model. (b) **Citation Authority**: The distribution of citations classified by source reliability.

metrics. We observe a strong positive correlation between human and LLM scoring (Total Score $r = 0.76$), which **validates the reliability of the automated evaluation framework** used in our main experiments. **(3) Robust Inter-Rater Agreement:** The consistently high $\alpha$ values in Figure 5(b), particularly in the Presentation dimension ($\alpha = 0.86$), underscore that the visual and structural advantages of FinSight are objectively recognizable and consensus-based.

## 4 ANALYSIS

### 4.1 RELIABILITY AND FACTUALITY ANALYSIS

**Key Fact Accuracy.** Directly measuring the factuality of long-form reports is challenging. We introduced a **Golden Facts Evaluation** by extracting 13 core financial indicators (e.g., Gross Margin, ROE) from professional reports as Ground Truth. We manually verified the recall rate of these data points. As shown in Figure 6(a), FinSight achieves a recall rate of **54.6%**, surpassing Gemini DR (38.5%) and OpenAI DR (30.0%) by a large margin. This demonstrates our method's superiority in uncovering deep, quantitative details that general-purpose agents often miss.

**Citation Quality Analysis.** We further evaluated the quality of the references cited in our report. (1) **Citation Faithfulness:** We manually verified the top-50 citations per report to check if the source explicitly supported the text. FinSight achieves a superior accuracy of **72.9%** (342/469 verified), outperforming Gemini DR's **69.8%**. This high faithfulness is attributed to our generative retrieval mechanism, which identifies references during the drafting process rather than via post-hoc appending. (2)**Source Authority:** We classified citations into High, Medium, and Low authority based on human-written rules. Figure 6(b) reveals that FinSight utilizes High Authority sources at a rate (36.5%) comparable to Gemini DR. While our reliance on open web search results in a slightly higher portion of Low Authority sources, the high faithfulness score ensures that the information extracted remains valid.

Table 3: Statistics of our generation process. We analyze metrics at both the CoA level and the final report level.

| Metric | Avg. Value |
| --- | --- |
| ***Chain of Analysis (CoA)*** | |
| # Tokens | 2,761 |
| # Images | 5.3 |
| ***Final Report*** | |
| # Fin. API Calls | 18.3 |
| # Search Queries | 983.2 |
| # Browse Pages | 469.8 |
| # CoA Segments | 17.6 |
| # Tokens | 62,586 |
| # Images | 51.2 |

### 4.2 STATISTICAL ANALYSIS OF GENERATION PROCESS.

Table 3 summarizes report statistics. Some key findings are: (1) Each CoA is a self-contained multimodal block, averaging 2,761 tokens and 5.3 images, (2) A report synthesizes about 17.6 CoAs, yielding 62,586 tokens and 51.2 images and (3) Incorporating deep search introduces richer knowledge, with 983.2 searches and 469.8 browsed pages per report.

### 4.3 ANALYSIS OF IMAGE GENERATION.

As illustrated in Figure 4, our Iterative Vision-Enhanced Mechanism progressively refines a stock chart over three iterations. In contrast to the initial, simplistic plot with low information density, the final visualization resolves this issue by integrating price and volume on a dual-axis, enriched with analytical overlays and contextual event markers, thereby presenting multifaceted data within a single view. This process is driven by critical VLM feedback across iterations, which targets improvements in aesthetics, information density, and other aspects.This suggests our mechanism is crucial for bridging the gap between automated chart generation and expert-quality financial visualizations.

## 5 RELATED WORK

### 5.1 DEEP RESEARCH SYSTEMS

Deep research systems represent a paradigm shift from traditional information retrieval to comprehensive knowledge synthesis, characterized by their ability to conduct multi-round information searching and integration. Current open-source deep research frameworks have emerged along several technical trajectories. ReAct-based agents (Yao et al., 2022), such as Open Deep Research (OpenAI, 2025a) and WebThinker (Li et al., 2025b), employ observation-thought-action loops with reasoning capabilities for iterative problem planning and execution. Multi-agent systems, including OWL (Hu et al., 2025) and Auto Deep Research (Tang et al., 2025), focus on collaborative problem-solving through agent specialization and coordination. Additionally, commercial systems represented by OpenAI Deep Research (OpenAI, 2025a) have demonstrated promising performance. However, existing frameworks exhibit significant limitations in multimodal processing (Yang et al., 2025) and domain-specific applications (Jimeno-Yepes et al., 2024; Tian et al., 2025). Due to the text-centric design of report generation workflows and the base models' lack of native image generation capabilities (Ren et al., 2021; Chen et al., 2024), current systems produce reports deficient in visual elements such as charts and diagrams. Furthermore, these systems demonstrate inadequate adaptation to financial domains, particularly in their inability to support for professional-grade chart generation, limited real-time market data integration, creating substantial gaps between system outputs and professional requirements.

### 5.2 LLM AGENTS IN FINANCIAL DOMAIN

Recent advances in Large Language Models have led to the development of various financial AI systems, each targeting specific aspects of financial analysis. Many of these works focus on stock price prediction and modeling (Zhang et al., 2025; Xiao et al., 2025) using multi-agent architectures. From a report generation perspective, FinTeam (Wu et al., 2025) can provide analysis from multiple viewpoints including company and industry levels. However, due to its single-round generation process, the resulting analysis lacks depth and comprehensiveness. Similarly, FinRobot (Yang et al., 2024) directly inputs collected information to models for single-round investment recommendation generation. Additionally, several open-source works (Zhang et al., 2025; Tian et al., 2025) provide comprehensive tools and data interfaces, yet they lack well-designed frameworks for report generation. Overall, existing systems exhibit critical limitations for comprehensive financial research report generation, particularly regarding report depth, data breadth, and multimodal integration.

## 6 CONCLUSION

In this paper, we present FinSight, a multi-agent framework designed for multimodal deep research. To the best of our knowledge, this is the first work capable of generating comprehensive, long-form reports with interleaved text and images. By integrating the Code Agent with Variable Memory and an Iterative Vision-Enhanced Mechanism, FinSight achieves dynamic analysis and professional visualization. While benchmarked on financial tasks, our code-centric approach provides a promising paradigm for future general-purpose automated research.

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

APPENDIX

Table 4: Overall evaluation results with 95% confidence intervals (subscript). **Bold** denotes the highest score, Underlined denotes the second highest.

| Model | Factual | | | Analytical | | | Presentation | | | Avg. |
|---|---|---|---|---|---|---|---|---|---|---|
| | Cons. | Faith. | T-I. | Rich. | Cover. | Ins. | Logic | Lang. | Vis. | |
| *LLM with Search Tools* | | | | | | | | | | |
| GPT-5 w/ Search | $5.95_{\pm0.48}$ | $6.35_{\pm0.39}$ | $4.77_{\pm0.72}$ | $5.43_{\pm0.58}$ | $4.52_{\pm0.51}$ | $5.09_{\pm0.49}$ | $6.53_{\pm0.27}$ | $5.87_{\pm0.35}$ | $3.90_{\pm0.79}$ | $5.38_{\pm0.51}$ |
| Claude-4.1-Sonnet | $5.78_{\pm0.52}$ | $5.92_{\pm0.55}$ | $\underline{3.55}_{\pm0.65}$ | $5.58_{\pm0.49}$ | $5.25_{\pm0.51}$ | $5.01_{\pm0.46}$ | $6.34_{\pm0.26}$ | $6.07_{\pm0.40}$ | $\underline{2.59}_{\pm0.51}$ | $5.12_{\pm0.48}$ |
| DeepSeek-R1 | $6.26_{\pm0.36}$ | $5.92_{\pm0.44}$ | $4.08_{\pm0.37}$ | $6.68_{\pm0.26}$ | $6.33_{\pm0.34}$ | $6.62_{\pm0.41}$ | $7.03_{\pm0.20}$ | $6.79_{\pm0.29}$ | $3.35_{\pm0.45}$ | $5.90_{\pm0.35}$ |
| *Deep Research Agent* | | | | | | | | | | |
| Grok Deep | $4.71_{\pm0.81}$ | $5.72_{\pm0.59}$ | $4.21_{\pm0.57}$ | $4.90_{\pm0.55}$ | $4.03_{\pm0.58}$ | $4.35_{\pm0.55}$ | $5.87_{\pm0.42}$ | $5.61_{\pm0.48}$ | $3.76_{\pm0.53}$ | $4.79_{\pm0.56}$ |
| Perplexity Deep | $5.02_{\pm0.61}$ | $5.74_{\pm0.63}$ | $4.03_{\pm0.97}$ | $3.88_{\pm0.60}$ | $3.40_{\pm0.53}$ | $3.65_{\pm0.59}$ | $5.47_{\pm0.44}$ | $4.92_{\pm0.47}$ | $3.42_{\pm0.97}$ | $4.39_{\pm0.65}$ |
| Gemini-2.5-Pro | $5.92_{\pm0.61}$ | $6.66_{\pm0.56}$ | $4.32_{\pm0.91}$ | $6.19_{\pm0.65}$ | $6.03_{\pm0.61}$ | $5.74_{\pm0.62}$ | $6.77_{\pm0.40}$ | $6.70_{\pm0.43}$ | $3.23_{\pm0.95}$ | $5.73_{\pm0.64}$ |
| OpenAI Deep | $\mathbf{6.87}_{\pm0.52}$ | $6.78_{\pm0.34}$ | $4.58_{\pm0.51}$ | $6.79_{\pm0.35}$ | $6.83_{\pm0.30}$ | $7.33_{\pm0.40}$ | $7.56_{\pm0.25}$ | $7.58_{\pm0.31}$ | $3.66_{\pm0.50}$ | $6.44_{\pm0.39}$ |
| **FinSight (ours)** | $\underline{6.84}_{\pm0.59}$ | $\mathbf{7.59}_{\pm0.52}$ | $\mathbf{7.84}_{\pm0.52}$ | $\mathbf{8.49}_{\pm0.47}$ | $\mathbf{8.44}_{\pm0.50}$ | $\mathbf{7.78}_{\pm0.48}$ | $\mathbf{7.82}_{\pm0.38}$ | $\mathbf{7.98}_{\pm0.33}$ | $\mathbf{8.57}_{\pm0.57}$ | $\mathbf{7.93}_{\pm0.49}$ |

## A    STATEMENT ON THE USE OF LARGE LANGUAGE MODELS (LLMS)

During the preparation of this manuscript, we use Large Language Models (LLMs) as a general-purpose assistance tool. The primary role of the LLM is to aid in improving the clarity and readability of the text, as well as to accelerate the implementation of our research ideas. Specific applications include: (1) Language and Grammar Correction: Polishing sentence structure, correcting grammatical erros, and refining word choices to enhance the overall quality of the writing. (2) Paraphrasing and Style Refinement: Rephrasing sentences and paragraphs to ensure consistency in tone and style throughout the paper. (3) Code Implementation Assistance: Generating code snippets and providing debugging support to help implement the proposed algorithms and experimental setups.

It should be noted that all core research concepts, experimental design, data analysis, and conclusions are developed exclusively by the human authors. Any content or suggestions generated by the LLM, including code, are critically checked, and substantially edited by the authors to ensure accuracy. The authors take full responsibility for the final content of this paper.

## B    FURTHER ANALYSIS

### B.1    STABILITY ANALYSIS AND CONFIDENCE INTERVALS

To rigorously validate our LLM-based evaluation framework, we conducted a comprehensive stability analysis covering three aspects: (1) confidence intervals across repeated runs, (2) dimension-specific variance analysis, and (3) cross-model verification for bias mitigation.

#### B.1.1    CONFIDENCE INTERVALS ACROSS REPEATED RUNS

To measure the sensitivity of our evaluation to stochastic variations, we repeated the evaluation process three times using `Gemini-2.5-Pro` as the judge model. For each run, we use the same evaluation prompt and rubric but with different random seeds. The 95% Confidence Intervals (CI) are calculated as $\mu \pm 1.96 \times \sigma/\sqrt{n}$ where $n = 3$.

As shown in Table 4, the 95% confidence intervals are consistently within $\pm 1.0$ point across all metrics, demonstrating the stability of our evaluation protocol. We attribute this to the robust evaluation design that anchors scoring against a provided "Golden Report" and utilizes a detailed, list-wise grading rubric.

#### B.1.2    DIMENSION-SPECIFIC VARIANCE ANALYSIS

We further analyzed the standard deviation across specific evaluation dimensions to understand which aspects of report quality are more reliably assessed by LLM judges.

Table 5: Average standard deviation across all models for each evaluation dimension.

| Dimension | Avg. Std. | Stability Level |
|---|---|---|
| Structural Logic (Logic) | 0.675 | High |
| Professional Language (Lang.) | 0.848 | High |
| Analytical Insight (Ins.) | 0.932 | High |
| Information Coverage (Cover.) | 0.966 | High |
| Core Conclusion Consistency (Cons.) | 1.010 | Medium |
| Information Richness (Rich.) | 1.058 | Medium |
| Textual Faithfulness (Faith.) | 1.213 | Medium |
| Chart Expressiveness (Vis.) | 1.340 | Lower |
| Text-Image Coherence (T-I.) | 1.470 | Lower |

Table 5 reveals important patterns: (1) **Structural and linguistic metrics** (e.g., *Structural Logic*, *Analytical Insight*) exhibit high stability with Std $< 1.0$, indicating that LLM judges reliably assess writing quality. (2) **Factual metrics** (e.g., *Textual Faithfulness*) show moderate variance, reflecting inherent difficulty in verifying factual claims. (3) **Visual metrics** (e.g., *Text-Image Coherence*, *Chart Expressiveness*) display slightly higher variance, as multimodal assessment involves more subjective judgment. Despite these variations, the overall ranking of methods remains consistent across runs.

### B.1.3 CROSS-MODEL VERIFICATION FOR BIAS MITIGATION

To investigate whether `Gemini-2.5-Pro` exhibited "self-preference bias" (favoring its own outputs), we employed `GPT-5` as an independent evaluator using the identical prompt and rubric.

Table 6: Cross-model evaluation comparison between GPT-5 and Gemini-2.5-Pro as judges.

| Method | Score (GPT-5) | Score (Gemini) | Rank (GPT-5) | Rank (Gemini) |
|---|---|---|---|---|
| GPT-5 w/ Search | 6.63 | 5.38 | 4 | 5 |
| Claude-4.1-Sonnet w/ Search | 4.71 | 5.12 | 7 | 6 |
| DeepSeek-R1 w/ Search | 6.75 | 5.90 | 3 | 3 |
| Grok Deep Search | 4.71 | 4.79 | 8 | 7 |
| Perplexity Deep Research | 6.08 | 4.39 | 6 | 8 |
| OpenAI Deep Research | 6.19 | 5.73 | 5 | 4 |
| Gemini-2.5-Pro Deep Research | 6.85 | 6.44 | 2 | 2 |
| **FinSight (ours)** | **8.04** | **7.93** | **1** | **1** |

As shown in Table 6, the ranking order remains highly consistent between the two judges (Kendall's $\tau = 0.764$, $p = 0.008$). Notably, even when evaluated by GPT-5, Gemini-2.5-Pro Deep Research retains the second-place position, and FinSight consistently achieves the top rank. This confirms that FinSight's superior performance is attributable to objective report quality rather than evaluator bias.

### B.2 QUANTITATIVE VISUAL ANALYSIS

To rigorously quantify the **presentation quality** and validate the effectiveness of our **Iterative Vision-Enhanced Mechanism**, we implement three reference-free Image Quality Assessment (IQA) metrics Hasler & Süsstrunk (2003):

- **Colorfulness**: Measures the chromatic distinction between visual elements, computed as $\sqrt{\sigma_{rg}^2 + \sigma_{yb}^2} + 0.3 \times \sqrt{\mu_{rg}^2 + \mu_{yb}^2}$, where $rg = R - G$ and $yb = 0.5(R + G) - B$.

- **RMS Contrast**: Measures luminance contrast using the root-mean-square of pixel intensities, correlating with the legibility of labels and grid lines.

- **Edge Density**: Measures information density versus visual clutter using Canny edge detection, computed as the ratio of edge pixels to total pixels.

We argue that pixel-wise metrics (e.g., MSE, SSIM) are ill-suited for chart evaluation, as different rendering engines produce large pixel discrepancies even when plotting identical data. The IQA metrics above provide a more meaningful assessment of visual quality.

Table 7: Quantitative visual quality comparison. Higher Colorfulness and Contrast indicate better aesthetic quality; Edge Density reflects information density.

| Method | Colorfulness | Contrast | Edge Density |
|---|---|---|---|
| **FinSight (Full)** | **32.35** | **31.71** | **0.0056** |
| *w/o Iterative Vision Mechanism* | 15.81 | 15.47 | 0.0027 |

As shown in Table 7, our Iterative Vision-Enhanced Mechanism approximately doubles the scores across all three dimensions, objectively validating that the VLM critic loop significantly improves the aesthetic quality and information density of the generated charts.

### B.3 ANALYSIS OF REPORT LENGTH AND QUALITY

To further investigate the characteristics of the generated reports, we analyze the relationship between report length and overall quality score, as illustrated in Figure 7. The plot shows that the outputs from our method are concentrated in the top-right quadrant, which indicates that our generated reports are not only comprehensive and of substantial length (typically over 20,000 words) but also of superior quality. We attribute this strong and consistent performance to our proposed two-stage writing framework. By first generating a concise Chain-of-Analysis, the model can then compose the final report based on richer, well-structured information, ensuring both analytical depth and coherence.

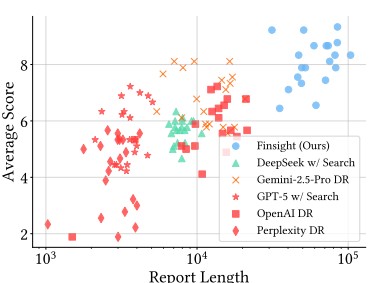

Figure 7: Correlation between report length and quality score across different methods.

In contrast, baseline methods exhibit significant limitations. Simpler approaches like LLM with search tool, which often rely on single-pass generation, are typically constrained to shorter reports. Meanwhile, other deep research agents such as OpenAI DR and Perplexity DR display a wide scatter of data points across the plot, which signifies a critical lack of consistency. For these methods, a greater length does not reliably translate into higher quality, highlighting the effectiveness of our structured, two-stage approach.

**Length-Controlled Evaluation.** To address potential length bias in LLM-based evaluations, we conducted an additional experiment with strict length constraints. We applied a truncation strategy to limit FinSight's output to approximately 10,000 words, aligning it with baseline models.

Table 8: Length-controlled evaluation results. FinSight (10k) represents reports truncated to 10,000 words.

| Method | Cons. | Faith. | T-I. | Rich. | Cover. | Ins. | Logic | Lang. | Vis. | Avg. |
|---|---|---|---|---|---|---|---|---|---|---|
| OpenAI DR | 5.60 | 7.45 | 4.90 | 6.35 | 6.40 | 5.90 | 6.90 | 6.85 | 4.65 | 6.11 |
| Gemini-2.5-Pro DR | 7.10 | 6.80 | 4.65 | 7.45 | 7.75 | 7.85 | 7.65 | 7.85 | 4.25 | 6.82 |
| FinSight (Full) | 6.85 | 7.50 | 7.85 | 8.70 | 8.30 | 8.45 | 8.05 | 8.10 | 9.00 | **8.09** |
| FinSight (10k) | 6.20 | 6.70 | 7.00 | 6.60 | 6.75 | 7.05 | 6.60 | 7.50 | 8.05 | 6.93 |

As shown in Table 8, even with forced truncation (which naturally penalizes coherence and completeness), FinSight (10k) achieves an overall score of 6.93, surpassing Gemini-2.5-Pro (6.82) and significantly outperforming OpenAI Deep Research (6.11). This demonstrates that FinSight's performance gain derives from high-quality content synthesis rather than mere verbosity.

# C   IMPLEMENTATION DETAILS

## C.1   BASELINES DETAILS

We mainly compare our method with the following two types of baselines:

**(1) LLMs with Search Tools.**

- **OpenAI GPT-5 w/Search**: The latest OpenAI's GPT model with web search API for research question.
- **Claude-4.1-Sonnet w/Search** The latest Anthropic's reasoning LLM with web search API for research question.
- **DeepSeek-R1 w/ Search**: The DeepSeek's LLM integrated with web search API for research question.

**(2) Deep Research Agents.**

- **Grok Deep Search**: The xAI's Deep Search applications, powered by the latest Grok model.
- **Perplexity Deep Research**: A commercial AI research assistant integrating multi-step search and analysis, optimized for rapid information aggregation.
- **OpenAI Deep Research**: A multi-step web research agent built on ChatGPT that searches, analyzes, and synthesizes information from multiple sources to produce research-grade reports with citations.
- **Gemini-2.5-Pro Deep Research**: Google's advanced research agent featuring multi-turn planning, deep web navigation, and multi-source evidence integration.

We evaluate these baselines directly on their official API and web applications. For consistency across different systems, we use the following unified prompt template to get the report. As commercial "black-box" systems, we have no control over their internal search routing or region settings. For our method, we set the Google Search API region to "China" to align with the benchmark dataset (A-share and HK-stock companies). We utilized English prompts for task instructions across all models, while keeping entity names (e.g., company names) in their original Chinese characters to ensure correct query interpretation.

We argue this setting is fair and potentially disadvantageous to FinSight for two reasons: (1) **Relevance over Bias**: Retrieval quality is driven primarily by query language and specificity rather than region settings. The China region setting was necessary to retrieve specific local filings. (2) **Commercial Advantage**: Commercial deep research systems often have access to high-quality search resources and curated financial databases with sophisticated internal query rewriting. In contrast, FinSight relies solely on the open Google Search API.

> **PROMPT**
>
> Please help me write a detailed research report on the corporate finance of {topic}, which should be rich in both text and charts. Give me the standardized citations at the end of the report (including serial numbers and corresponding references).

## C.2   DETAILS OF FINSIGHT

**Backbone**   For Multi-source Data Collection Agent, Deep Search Agent and Data Analysis Agent, we use the `DeepSeek-V3` as the backbone model. For Report Generation Agent, we use `DeepSeek-R1` as the backbone model. The maximum input length is 81,920 tokens, and the maximum output length is 16,384 tokens.

**Data Collection**   We implement the financial api tool based on akshare [2] package in Python. For web search, we use the Google Search API, with the region set to China and the number of retrieved

---

[2]https://github.com/akfamily/akshare

results fixed at the top 10. For web content acquisition, we employ Playwright [3] to simulate a browser for webpage content extraction.

**Retrieval**  We use `Qwen3-Embedding-0.6B` to generate embeddings for data and CoA segments. Then we use the cosine similarity to select the relevant data and CoA segments for each section.

**Iterative Vision-Enhanced Mechanism**  We use the `Qwen2.5-VL-72B` as the critic vision-language model in the chart generation stage. To balance effectiveness and cost, we perform three iterations of the critic process.

**Ablation Study**  We conduct ablation study on 5 company questions, which includes: Cambricon Technologies, Li Auto-W, Pop Mart, 3SBio, China Mobile. Some variants are as follows:

- **w/o Iteration Vision-Enhanced Mechanism** We remove the iterative refinement process and plot charts in a single pass.
- **w/o Two-Stage Writing Framework** We only concatenate the CoA segments to output the final report.
- **w/o Dynamic Search-Enhanced Strategy** We remove the Dynamic Search-Enhanced Strategy from the Data Collection and Report Generation process.

### C.3  CITATION ACCURACY EVALUATION

To rigorously evaluate the faithfulness of generated citations, we conducted a comprehensive manual verification study.

**Methodology.**  Human experts checked the top 50 citations in each generated report to verify whether the cited source actually supported the generated claim. For each citation, annotators classified it as *Accurate* (the source directly supports the claim), *Partially Accurate* (the source is related but does not fully support the claim), or *Inaccurate* (the source is irrelevant or contradicts the claim). We report the overall accuracy as the proportion of Accurate citations.

Table 9: Citation verification results across all company-level and industry-level tasks.

| Metric | FinSight | Gemini-2.5-Pro DR |
|---|---|---|
| Total Citations Checked | 469 | 414 |
| Overall Accuracy | **72.92%** (342/469) | 69.81% (289/414) |

As shown in Table 9, even while generating a higher volume of citations, FinSight maintains higher accuracy. We attribute this to our **Two-Stage Writing Framework** and generative retrieval mechanism, which identifies references during the drafting process rather than via post-hoc appending.

**Citation Authority Analysis.** We further analyzed source quality by classifying citations into three authority levels:

- **High Authority**: Government/Regulatory bodies (SEC, IMF), Official Company Filings, Top Academic/Research Institutions.
- **Medium Authority**: Mainstream Financial Media (Bloomberg, Reuters), Known Market Research Firms.
- **Low Authority**: Social Media, Personal Blogs, Content Farms, or unverified aggregators.

FinSight utilizes High Authority sources at a rate comparable to commercial baselines ($\sim$36.5%). The slightly higher usage of Low Authority sources compared to OpenAI DR reflects our reliance on open web search versus proprietary filtered databases, indicating a direction for future refinement.

---

[3]https://playwright.dev/

Table 10: Distribution of citation authority across different methods.

| Model | High | Medium | Low | Total |
|---|---|---|---|---|
| Gemini-2.5-Pro DR | 312 (36.7%) | 304 (35.8%) | 233 (27.4%) | 849 |
| OpenAI DR | 348 (35.0%) | 504 (50.7%) | 142 (14.3%) | 994 |
| FinSight | **334 (36.5%)** | 319 (34.8%) | 263 (28.7%) | 916 |

## C.4 KEY FACT RECALL EVALUATION

Directly measuring the factuality of long reports is challenging due to the absence of strict ground truth. To quantify factual accuracy, we introduced a **Golden Facts Evaluation** methodology.

**Methodology.** We extracted **13 core financial indicators** from the professional Golden Reports as ground truth, covering: (1) Profitability (Gross Margin, Net Margin), (2) Growth (Revenue Growth, Profit Growth), (3) Financial Health (Cash Flow, Debt Ratio), (4) Valuation (PE Ratio, PB Ratio), and (5) Efficiency (ROE, ROA). Human annotators then manually verified how many of these specific data points were accurately retrieved and reported by each model across company-level tasks.

Table 11: Key financial information recall across methods.

| Method | Avg. Hits (out of 13) | Recall Rate | Relative |
|---|---|---|---|
| FinSight | **7.1** | **54.6%** | – |
| Gemini-2.5-Pro DR | 5.0 | 38.5% | -29.6% |
| OpenAI DR | 3.9 | 30.0% | -45.1% |

FinSight achieves significantly higher recall rate, demonstrating superior coverage of critical financial data compared to commercial deep research systems.

## C.5 CONSTRUCTION OF THE FINANCIAL REPORT GENERATION BENCHMARK

**Questions** We select the most popular five A-share companies, five Hong Kong-stock companies, and ten representative industries from https://www.djyanbao.com as the benchmark research questions. These companies and industries cover a diverse set of market sectors and provide a comprehensive foundation for evaluating the effectiveness of deep research systems.

**Golden Referenced Report** To establish human expert-level benchmark, we collect the latest equity and industry research reports from well-known Chinese securities firms, as shown in Table 12. These golden references cover both company-level and industry-level analyses across A-shares, Hong Kong stocks, and major industries.

## C.6 EVALUATION METRICS

We further illustrate the metrics we used for evaluation:

**(1) Factual Metrics** Measure the textual quality and factual accuracy of the final report.

- **Core Conclusion Consistency**: Whether the core conclusions in the generated report are consistent with those in the reference report.

- **Textual Faithfulness**: Whether the arguments in the report are properly supported by citations from the reference.

- **Text-Image Coherence**: Whether the report integrates images into the discussion, and whether the textual and visual descriptions align.

Table 12: Golden Referenced Reports from Chinese Securities Firms

| Market | Company / Industry | Securities Firm |
|---|---|---|
| A-shares | SMIC (688981) | Soochow Securities |
| | Cambricon Technologies (688256) | Donghai Securities |
| | China Mobile (600941) | Zhongtai Securities |
| | Skshu Paint (603737) | Huatai Securities |
| | Yiwu China Commodities City (600415) | Guolian Minsheng Securities |
| Hong Kong Stocks | Pop Mart (09992) | Zhongtai Securities |
| | SenseTime (00020) | Zhongtai Securities |
| | Li Auto-W (02015) | Huayuan Securities |
| | 3SBio (01530) | Huatai Securities |
| | UBTECH Robotics (09880) | Guohai Securities |
| Industries | Semiconductor Industry | Kaiyuan Securities |
| | Food & Beverage Industry | Huachuang Securities |
| | Basic Chemical Industry | Zhongtai Securities |
| | Steel Industry | Orient Securities |
| | Construction & Decoration Industry | Guosheng Securities |
| | Environmental Protection & Public Utilities (Controlled Nuclear Fusion) | Huachuang Securities |
| | Light Manufacturing (Durable Consumer Goods) | Guotai Haitong Securities |
| | K12 Education Industry | Guosheng Securities |
| | Media Industry (Short Drama Overseas Expansion) | Soochow Securities |
| | Transportation (Cross-border E-commerce Logistics) | Maigao Securities |

**(2) Analysis Effectiveness** Measure whether the financial report provides sufficient information and insights for investors.

- **Information Richness**: The number of distinct information points included in the report.

- **Coverage**: The extent to which key information from the golden reference report is covered.

- **Analytical Insight**: Whether the report provides critical analysis, original insights, and forward-looking recommendations.

**(3) Presentation Quality** Measure the presentation quality of the final report.

- **Structural Logic**: The logical organization of each section and the overall structural soundness of the report.

- **Language Professionalism**: Whether the language conforms to financial terminology, using the golden report as a reference.

- **Chart Expressiveness**: The effectiveness of charts in supporting the narrative, including their informativeness and aesthetic quality.

C.7 LLM EVALUATION PROCESS

We adopt `Gemini-2.5-Pro` as the backbone evaluation model. To ensure fair comparison across reports, we employ a list-wise evaluation strategy, where the model is provided with all candidate reports along with the golden reference report and assigns scores accordingly. The nine metrics mentioned above can be divided into two parts, one is unrelated to the golden report and the other is related to the golden report. For these two types, we have designed two types of prompts, which are listed below.

**Evaluation Instruction for Golden Report Irrelevant Metrics**

# [TASK]

Your task is to act as an expert financial analyst and editor. You will perform a rigorous, **comparative evaluation** of a list of financial research reports. Your goal is to produce a structured critique for each report based on how effectively it addresses the central **Research Question**, using the provided **Golden Standard Report** as a quality benchmark.

# [INPUTS]

* **Research Question:** Research Question * **Golden Standard Report:** Given in file format, the one starting with 'golden' is the 'golden standard report' * **Reports to Evaluate:** Reports

# [EVALUATION METHODOLOGY]

To ensure fairness and accuracy, you must follow this three-step process for **each report** in the 'Reports to Evaluate' list:

1. **Step 1: Establish the Benchmark (Internal Thought Process)**

* For each of the six evaluation dimensions, first thoroughly analyze the **Golden Standard Report**. Identify its key characteristics, depth, and quality to create a mental benchmark for what constitutes a high-quality, professional report (which corresponds to a score of 7).

2. **Step 2: Comparative Analysis (Internal Thought Process)**

* Now, analyze the report currently being evaluated. For each dimension, find concrete evidence (e.g., specific quotes, data points, chart quality, structural features). * **Directly compare** this evidence against the benchmark established in Step 1. Note where the report meets, exceeds, or falls short of the Golden Standard.

3. **Step 3: Score and Justify (Final Output Generation)**

* Based on the comparison in Step 2, assign a score from 1 to 10 for the dimension, following the 'Benchmark-Based Scoring' rules below. * Write a **concise, one-sentence rationale** that justifies your score by referencing your comparative findings.

# [SCORING GUIDELINES]

Adhere strictly to these principles to maintain objectivity:

* **Benchmark-Based Scoring:**

* **The Golden Standard Report is the benchmark for a score of 7.** * A report demonstrating a **similar level of quality**, depth, and execution as the Golden Standard on a specific dimension should receive a score of **7**. * Scores of **8-10** are reserved for reports that **demonstrably exceed** the Golden Standard in that dimension (e.g., providing deeper insights, more comprehensive data, or superior visualizations). * Scores of **1-6** indicate that the report **falls short** of the Golden Standard's quality in that dimension, with the score reflecting the degree of the gap.

* **Justification for Extremes:** Scores of **9-10** (exceptional) or **1-2** (critically flawed) require a particularly strong and specific justification in the rationale.

# [EVALUATION FRAMEWORK and CRITERIA]

### **Dimension 1: Information Richness (Score 1-10)**

* **Definition:** Measures the concentration of substantive, verifiable facts and data points relevant to the research question, while minimizing filler content.

### **Dimension 2: Textual Faithfulness (Score 1-10)**

* **Definition:** Measures whether significant claims, data, and forecasts are verifiably supported by provided "References / Data Sources".

### **Dimension 3: Text-Image Coherence (Score 1-10)**

* **Definition:** Assesses if charts and tables are consistent with the text and if the text provides meaningful interpretation that supports the core analysis.

### **Dimension 4: Analytical Insight (Score 1-10)**

* **Definition:** Evaluates the quality of the analysis, focusing on critical thinking, original insights, and actionable, forward-looking conclusions that directly address the research question.

### **Dimension 5: Structural Logic (Score 1-10)**

* **Definition:** Measures the structural integrity and logical flow of the argument, assessing if the report builds a clear and compelling case from evidence to conclusion.

### **Dimension 6: Chart & Table Expressiveness (Score 1-10)**

* **Definition:** Focuses on the quality of data visualizations themselves—their clarity, ability to reveal patterns, and effectiveness in communicating key information.

# [OUTPUT FORMAT]

Provide your evaluation in the following strict JSON format. **For each score, you must provide a brief, one-sentence rationale.** Do not add any conversational text outside of this structure. Use the file name of each report as its report id.

Now start your evaluation of the given reports. Carefully read each report and give a score.

---

**Evaluation Instruction for Golden Report Relevant Metrics**

**[ROLE]** You are an expert financial analyst and editor, specializing in the comparative analysis of research reports.

**[TASK]** Your task is to rigorously evaluate a list of **Generated Reports** by comparing each one against a **Benchmark Report** (a professionally written 'gold standard'). You will assess each Generated Report's quality across three key dimensions on a scale of 1 to 10, producing a structured JSON output with scores and justifications.

**[INPUTS]**

1. **`Benchmark Report`**: A high-quality, professional research report that serves as the "gold standard" for this evaluation. All comparisons should be made against this document. The file name of the benchmark report begins with "golden_". 2. **`Generated Reports`**: A list of one or more reports to be evaluated against the Benchmark Report. 3. **`Report ID`**: An identifier for each Generated Report. Use the file name as the report ID.

**[EVALUATION METHODOLOGY]**

To ensure fairness and accuracy, you must follow this three-step process for **each Generated Report**:

1. **Step 1: Establish the Benchmark (Internal Thought Process)**
* For each of the three evaluation dimensions, first thoroughly analyze the **Benchmark Report**. Identify its key characteristics, depth, and quality to create a mental benchmark for what constitutes a score of **7**.

2. **Step 2: Comparative Analysis (Internal Thought Process)**
* Now, analyze the Generated Report. For each dimension, find concrete evidence (e.g., specific conclusions, data points included/omitted, linguistic style). * **Directly compare** this evidence against the benchmark established in Step 1. Note where the report meets, exceeds, or falls short of the Benchmark Report.

3. **Step 3: Score and Justify (Final Output Generation)**
* Based on the comparison in Step 2, assign a score from 1 to 10 for the dimension, following the `SCORING GUIDELINES` below. * Write a **concise, one-sentence rationale** that justifies your score by referencing your comparative findings.

**[SCORING GUIDELINES]**

Adhere strictly to these principles to maintain objectivity:

* **Benchmark-Based Scoring:** * **The Benchmark Report is the standard for a score of 7.** * A report demonstrating a **similar level of quality**, depth, and execution as the Benchmark Report on a specific dimension should receive a score of **7**. * Scores of **8-10** are reserved for reports that **demonstrably exceed** the Benchmark Report in that dimension (e.g., providing a more nuanced conclusion, broader data coverage, or more sophisticated language). * Scores of **1-6** indicate that the report **falls short** of the Benchmark Report's quality in that dimension, with the score reflecting the degree of the gap. * **Justification for Extremes:** Scores of **9-10** (exceptional) or **1-2** (critically flawed) require a particularly strong and specific justification in the rationale.

**[EVALUATION FRAMEWORK & CRITERIA]**

### **Dimension 1: Core Conclusion & Data Consistency (Score 1-10)**
* **Definition:** Measures the alignment of the Generated Report's core thesis, key arguments, and supporting data points with those presented in the Benchmark Report.

### **Dimension 2: Information Coverage (Score 1-10)**
* **Definition:** Assesses the extent to which the Generated Report includes the key information points, topics, and analytical angles present in the Benchmark Report.

### **Dimension 3: Professional Language & Tone (Score 1-10)**
* **Definition:** Evaluates the linguistic quality of the Generated Report, using the Benchmark Report's writing style, tone, and vocabulary as the standard for professional financial analysis.

**[OUTPUT FORMAT]** Provide your evaluation in the following strict JSON format. For each score, you must provide a brief, one-sentence rationale that explains the score relative to the benchmark. Do not add any conversational text outside of this structure.

Now start your evaluation of the given reports. Carefully read each report and give a score.

---

## C.8 HUMAN EVALUATION PROCESS

To validate our automated evaluation and substantiate claims about report quality, we conducted a comprehensive human evaluation study.

We recruited **6 graduate students with financial backgrounds** (majoring in Finance, Economics, or related fields) to serve as expert annotators. To save costs, we selected the two strongest baselines, **Gemini-2.5-Pro Deep Research** and **OpenAI Deep Research**, to compare against FinSight. Each annotator reviewed a random subset of 10 research topics, evaluating all three systems' outputs for each topic. In reviewing process, raters were provided with "Golden Reports" (professional analyst reports from top-tier securities firms) as ground truth references to anchor their judgments.

**Scoring Protocol.** To manage cognitive load when evaluating long-form reports, raters scored on a 0–5 scale with 0.5 increments across three consolidated dimensions: *Factual* (combining Consistency, Faithfulness, Text-Image Coherence), *Analytical* (combining Richness, Coverage, Insight), and *Presentation* (combining Logic, Language, Visualization). Scores were scaled ($\times 2$) to align with our 0–10 automated metrics.

Table 13: Human evaluation scores (scaled to 0-10).

| Model | Factual | Analytical | Presentation | Total |
|---|---|---|---|---|
| OpenAI DR | 5.93 | 5.81 | 4.75 | 5.50 |
| Gemini-2.5-Pro DR | **6.86** | 6.73 | 4.73 | 6.11 |
| FinSight | 6.68 | **7.17** | **7.48** | **7.11** |

Table 14: Human-LLM alignment and inter-rater reliability metrics.

| Dimension | Pearson $r$ | Krippendorff's $\alpha$ |
|---|---|---|
| Factual | 0.6360 | 0.4667 |
| Analytical | 0.6003 | 0.4752 |
| Presentation | 0.6757 | 0.8570 |
| **Total Score** | **0.7587** | **0.6474** |

We calculated Inter-Rater Reliability using Krippendorff's Alpha ($\alpha$) and Human-LLM Alignment using Pearson correlation coefficient ($r$). Our key findings are as follows: (1) FinSight achieves the highest total score (7.11), significantly outperforming both commercial baselines. (2) The strong positive correlation between human and LLM scoring (Pearson $r > 0.75$ for Total Score) validates the reliability of our automated evaluation framework. (3) The overall inter-rater reliability ($\alpha = 0.64$) indicates solid consensus among experts, with exceptionally high agreement in the Presentation dimension ($\alpha = 0.86$), confirming that FinSight's multimodal capabilities provide objectively recognizable advantages.

The instruction for human raters is as follows.

---

**Evaluation Instruction for Human Raters**

# General Instructions

Thank you for participating in this evaluation. Please assess each report independently based on the three core dimensions defined below. Each dimension is scored on a **1 to 5 point scale**, allowing for half-points (e.g., 3.5).

- **5 points (Excellent):** Significantly exceeds expectations; outstanding performance in all aspects.
- **4 points (Good):** Solid and reliable; comprehensively meets all requirements for a professional report.
- **3 points (Passable):** Fundamentally adequate, but with clear deficiencies in some areas.
- **2 points (Poor):** Contains serious flaws; fails to deliver core value.
- **1 point (Very Poor):** Contains almost no usable information; logically incoherent or factually incorrect.

---

# Dimension 1: Factual - Accuracy & Comprehensiveness

**Definition:** Assesses the **truthfulness, completeness,** and **objective evidence** of the information provided in the report. This dimension concerns the solidity of the report's foundation.

| Score | Evaluation Criteria |
| --- | --- |
| 5 (Excellent) | Information is extremely dense, facts are cross-verified and accurate, all key topics are covered, and crucial data is clearly supported by sources. |
| 4 (Good) | Information is solid, facts are generally accurate, most key topics are covered, and major data points are supported by sources. |
| 3 (Passable) | Contains basic facts, but coverage is insufficient (e.g., missing key information points), or there are minor factual errors / missing sources. |
| 2 (Poor) | Contains numerous factual errors or severe gaps in information; most claims are not supported by data or sources. |
| 1 (Very Poor) | Filled with unverified information, obvious factual errors, or large-scale content omissions. |

# Dimension 2: Analytical - Depth & Logic

**Definition:** Assesses the **quality of analysis, insightfulness,** and **argumentative structure** of the report. This dimension concerns whether the report provides added value beyond a simple recitation of facts.

| Score | Evaluation Criteria |
| --- | --- |
| 5 (Excellent) | Insights are profound, drawing unique and forward-looking conclusions from the data. The logical chain is complete, rigorous, and highly persuasive. |
| 4 (Good) | Analysis is reasonable and capable of effective deduction based on facts. The logic is clear, the structure is complete, and the conclusion is consistent with the argumentation. |
| 3 (Passable) | Contains basic analysis, but lacks depth (often just restating facts). The logic is generally coherent but not sufficiently rigorous. |
| 2 (Poor) | Analysis is superficial or contains logical leaps. There is a weak connection between arguments and evidence; the structure is chaotic. |
| 1 (Very Poor) | Almost no analysis, or filled with logical contradictions. Fails to form a coherent argument. |

# Dimension 3: Presentation - Quality & Professionalism

**Definition:** Assesses the **readability, effectiveness of charts,** and **professionalism of the language**. This dimension concerns whether the report can be understood efficiently and clearly.

| Score | Evaluation Criteria |
| --- | --- |
| 5 (Excellent) | Language is precise, professional, and authoritative. Charts are exceptionally well-designed, perfectly complementing the text and greatly enhancing the argument. |
| 4 (Good) | Language is professional and fluent. Charts are clear, easy to understand, and effectively support the text's points; figure-text consistency is good. |
| 3 (Passable) | Language is generally professional but occasionally verbose or inappropriate. Chart quality is average (e.g., unclear, low information), or the connection to the text is weak. |
| 2 (Poor) | Language is unprofessional or contains many errors. Chart quality is poor (e.g., misleading, unreadable), or there is a serious disconnect between figures and text. |
| 1 (Very Poor) | Language is confusing and difficult to read. No charts are used, or the charts provided are completely ineffective. |

# D A CASE OF COMPANY RESEARCH QUESTION

To demonstrate the practical application of our system, this section shows the case of **SenseTime Technology (0020.HK)**, a leading artificial intelligence company in China.

We present the collecting tasks of the Data Collection process in Table 15, and an analytical tasks of the Data Analysis process in Table 16.

Table 15: The predefined and brainstormed data collection tasks.

**Data Collection**

```
1 Predefined Tasks:
2 "company": [
3     {"name": "Balance Sheet"},
4     {"name": "Income Statement"},
5     {"name": "Cash Flow Statement"},
6     {"name": "Basic Stock Information"},
7     {"name": "Shareholder Structure"},
8     {"name": "Stock Price"},
9     {"name": "Stock-related Financial Data"},
10    {"name": "CSI 300 Daily Index Data"},
11    {"name": "Hang Seng Daily Index Data"},
12    {"name": "NASDAQ Daily Index Data"},
13    {"name": "Investment Rating",
14    "description": "Collect analyst investment ratings and target prices
       from major securities firms (e.g., CITIC Securities, CICC). Select
       the most frequent rating across institutions."
15    },
16    {"name": "Price-to-Sales Ratio (PS)",
17    "description": "Collect the company's PS data (PS = Stock Price /
       Sales per Share)."
18    },
19    {"name": "Return on Equity (ROE, %)",
20    "description": "Collect the company's ROE data, defined as (Net
       Income Attributable to Shareholders / Equity) or equivalently (Net
       Profit Margin x Asset Turnover x Equity Multiplier)."
21    },
22    { "name": "Competitor Data",
23    "description": "Collect comparable data from 5-8 major peer
       companies, including financial and operational indicators."
24    },
25 ]
26
27 Brainstormed Tasks: [
28    {"name": "Growth Data for AI Subfields in 2024"},
29    {"name": "The Development History of The SenseTime and Main Business
       "}
30 ]
```

Table 16: The illustration of data analysis process and CoA output of a task.

| **Data Analysis (We show a predefine task of data analysis, and its CoA output.)** |
| --- |

**Predefined Task 1:** Company Development History and Main Business Analysis
**Required Data:** Years and corresponding company achievements.
**Description:** Organize the company's key milestones by year (e.g., founding, expansions, product launches, strategic shifts, and M&A) and present them as a timeline with annotated events. Also summarize its current core business scope, including main products, business segments, and market positioning.

**CoA Output:**

## SenseTime: Evolution from Computer Vision Pioneer to AI Platform Enterprise
### Executive Summary
As a global leading artificial intelligence software company, SenseTime has grown into an AI giant valued at over USD 10 billion since its founding in 2014. With the mission of "persisting in originality and enabling AI to advance human progress", the company has built a comprehensive AI ecosystem covering four major business segments: Smart Business, Smart City, Smart Life, and Smart Auto, through 11 years of continuous innovation and business expansion [Source: SenseTime Group Company Profile]. By 2024, SenseTime employed 3,756 people, with operations spanning over 20 cities worldwide and R&D centers in Hong Kong, Shanghai, Beijing, and Shenzhen [Source: SenseTime Basic Information].

The company's growth trajectory reflects a typical path of technology firms: from technical breakthroughs, to productization, then to platformization and ecosystemization. Its IPO on the Hong Kong Stock Exchange in 2021 (Ticker: 0020.HK) marked the beginning of a new capital-intensive phase, while the launch of the "Riyadh New" large model system in 2023 signified its strategic pivot toward generative AI [Source: Development History Search Data]. Currently, SenseTime reports annual revenue of RMB 3.772 billion, maintaining technological leadership in computer vision and artificial intelligence [Source: Stock 0020 Income Statement Data].

### Development History Analysis
**Early Stage of Founding and Technology Accumulation (2014–2016)**
Founded on October 15, 2014, in the Cayman Islands by the CUHK Multimedia Lab team, SenseTime initially focused on core R&D in computer vision and deep learning [Source: SenseTime Basic Information]. In 2015, it launched its first facial recognition algorithm and completed its first funding round, laying the foundation for subsequent commercialization. By 2016, it had become the world's most valuable AI unicorn, reflecting strong capital market recognition.

@import "SenseTime Development Timeline (2014-2025), horizontal axis = year, vertical axis = milestone importance, annotated with key events (milestone_df)"

**Productization and Market Expansion (2017–2019)**
In 2017, the launch of the SenseAR augmented reality platform marked the transition from pure research to productization. By 2018, it had completed Series C financing with a valuation exceeding USD 4.5 billion. In 2019, it rolled out Smart City solutions, partnering with key domestic cities and expanding its scope from technology output to integrated solutions.
**Platformization and Capitalization (2020–2022)**
In 2020, the release of the SenseCore AI Infrastructure introduced the concept of industrializing AI, aiming to reduce the cost of AI production factors. On December 30, 2021, SenseTime was listed on the Hong Kong Stock Exchange Main Board, raising about HKD 6 billion [Source: SenseTime IPO Search Data]. In 2022, it launched the "Riyadh New" foundation model system, officially entering the generative AI space and advancing toward multimodal development.
**Ecosystemization and Deepening Applications (2023–2025)**
In 2023, the company released the "SenseChat" large language model, further completing its model matrix. In 2024, its Smart Auto business grew rapidly, with partnerships across major automakers. By 2025, SenseTime continued to optimize its large model technologies and achieved deep application deployments in healthcare, finance, and manufacturing.

### Core Business Structure Analysis

**Smart Business (35%)**

Core Products: SenseFoundry Enterprise Ark, AI Content Generation Platform Market Positioning: Provides AI solutions for enterprise digital transformation, focusing on finance, retail, and manufacturing. This is the main revenue driver, contributing about 35% of revenue in 2024 [Source: Business Segment Data].

**Smart City (30%)**

Core Products: City Ark, traffic management, and surveillance systems

Market Positioning: Provides smart governance and public service solutions. Established partnerships with major cities, gaining competitive advantages in smart transportation and public safety.

**Smart Life (20%)**

Core Products: SenseMARS Metaverse Platform, mobile AI applications

Market Positioning: Consumer-facing AI applications and metaverse ecosystem. Focuses on AR/VR and mobile internet scenarios, with strong growth potential.

**Smart Auto (15%)**

Core Products: SenseAuto Platform, autonomous driving solutions

Market Positioning: Provides intelligent driving and V2X systems for automakers. This segment grew rapidly in 2024, becoming a new growth engine [Source: Smart Auto Data].

**@import "SenseTime 2024 Revenue Breakdown Pie Chart, showing four business segments (business_df)"**

### Technical Strength and Competitive Advantages

SenseTime possesses deep academic roots and originality in AI technologies. The company has built full-stack AI capabilities across perception intelligence, decision intelligence, content generation, and content enhancement [Source: Company Profile].

By 2024, it had over 8,000 AI-related patents, maintaining leadership in computer vision, large model training, and AI chips [Source: Patent Data].

Core Competitive Advantages:

1. Technological Originality: Commitment to independent fundamental research and development
2. Platformization Capability: SenseCore enables AI industrial-scale production
3. Ecosystem Completeness: Full product portfolio covering four major domains
4. Talent Density: A world-class AI R&D team

### Market Position and Prospects

Industry data show SenseTime as a global leader in computer vision, ranking among top players in market share. In 2024, it was positioned in the Leaders Quadrant of the AI maturity curve, highly rated in both innovation and execution [Source: AI Market Growth Data 2024].

Future Directions:

- Large Model Enhancement: Continuous optimization of the "Riyadh New" large model system
- Vertical Integration: Deepened applications in finance, healthcare, and manufacturing
- International Expansion: Strengthening presence in Southeast Asia and the Middle East
- Tech Convergence: Driving integration of AI with IoT and blockchain

**@import "SenseTime Revenue Growth Trend (2020-2024), showing revenue growth from RMB 3.446B to RMB 3.772B (income_statement)"**

### Conclusion

Through 11 years of development, SenseTime has evolved from a single computer vision technology provider into a comprehensive AI platform enterprise. Its trajectory reflects a clear progression: technology-driven, product iteration, platform building, and ecosystem expansion. With balanced business segments and strong technological capabilities, SenseTime holds a favorable competitive position in the AI industrialization wave.

Looking forward, with continuous AI evolution and expanding applications, SenseTime is poised to play a more significant role in the global AI market. To remain competitive, the company must sustain R&D investment, deepen vertical applications, and enhance international collaborations in response to intensifying market and technological challenges.

# E   REPORT GALLERY

We have presented an overview of the report generated by  ours  here, and the complete report can be obtained from https://anonymous.4open.science/r/FinSight-6739/.

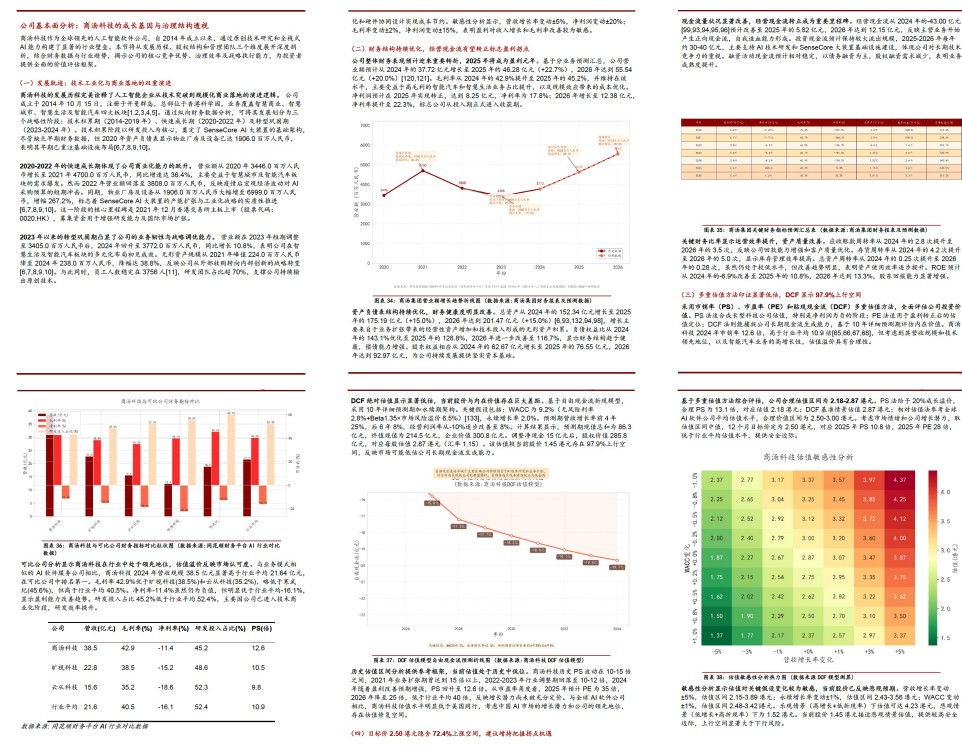

Figure 8: The final report of The SenseTime (part).

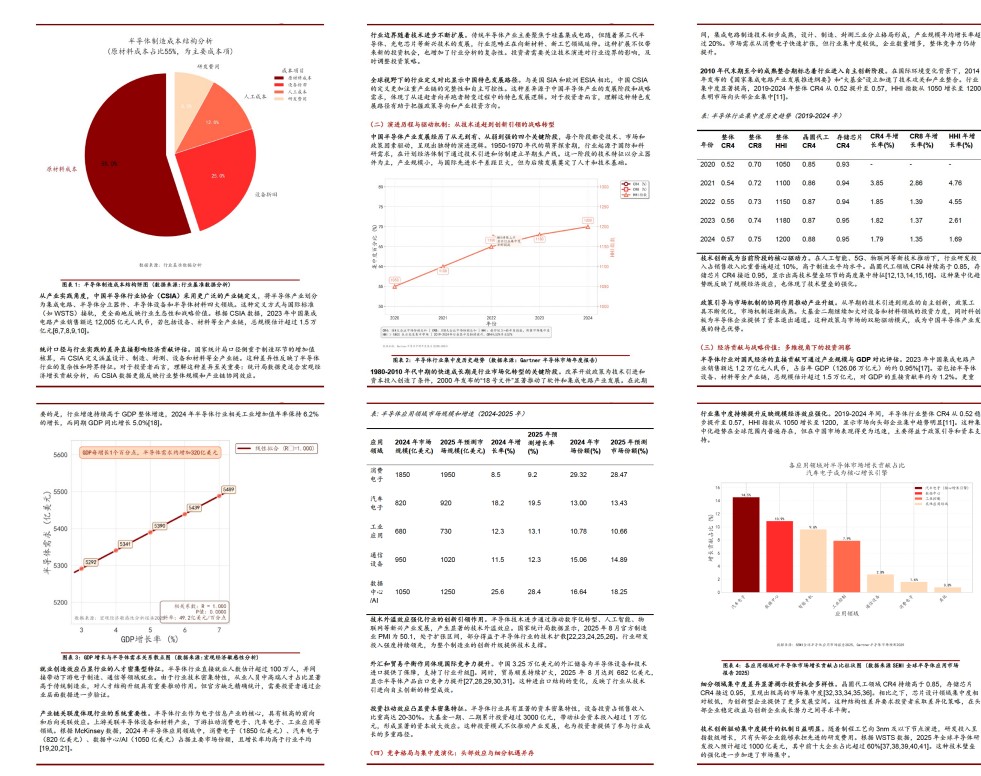

Figure 9: The final report of semiconductor industry (part).

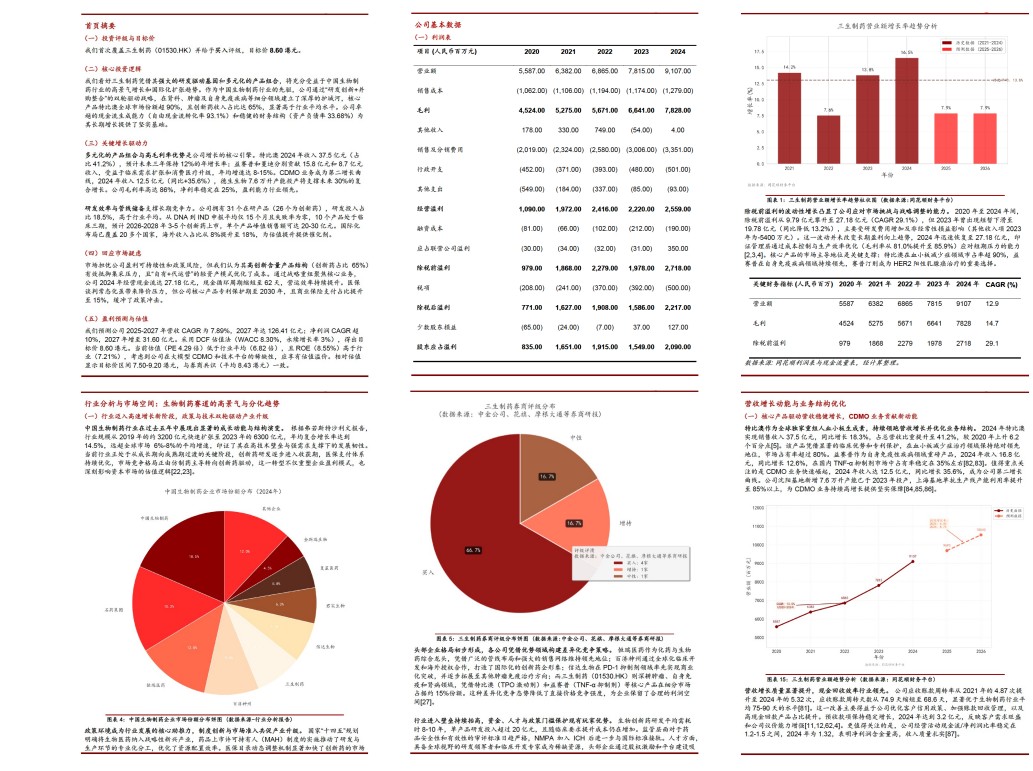

Figure 10: The final report of The 3SBio Inc. (part).

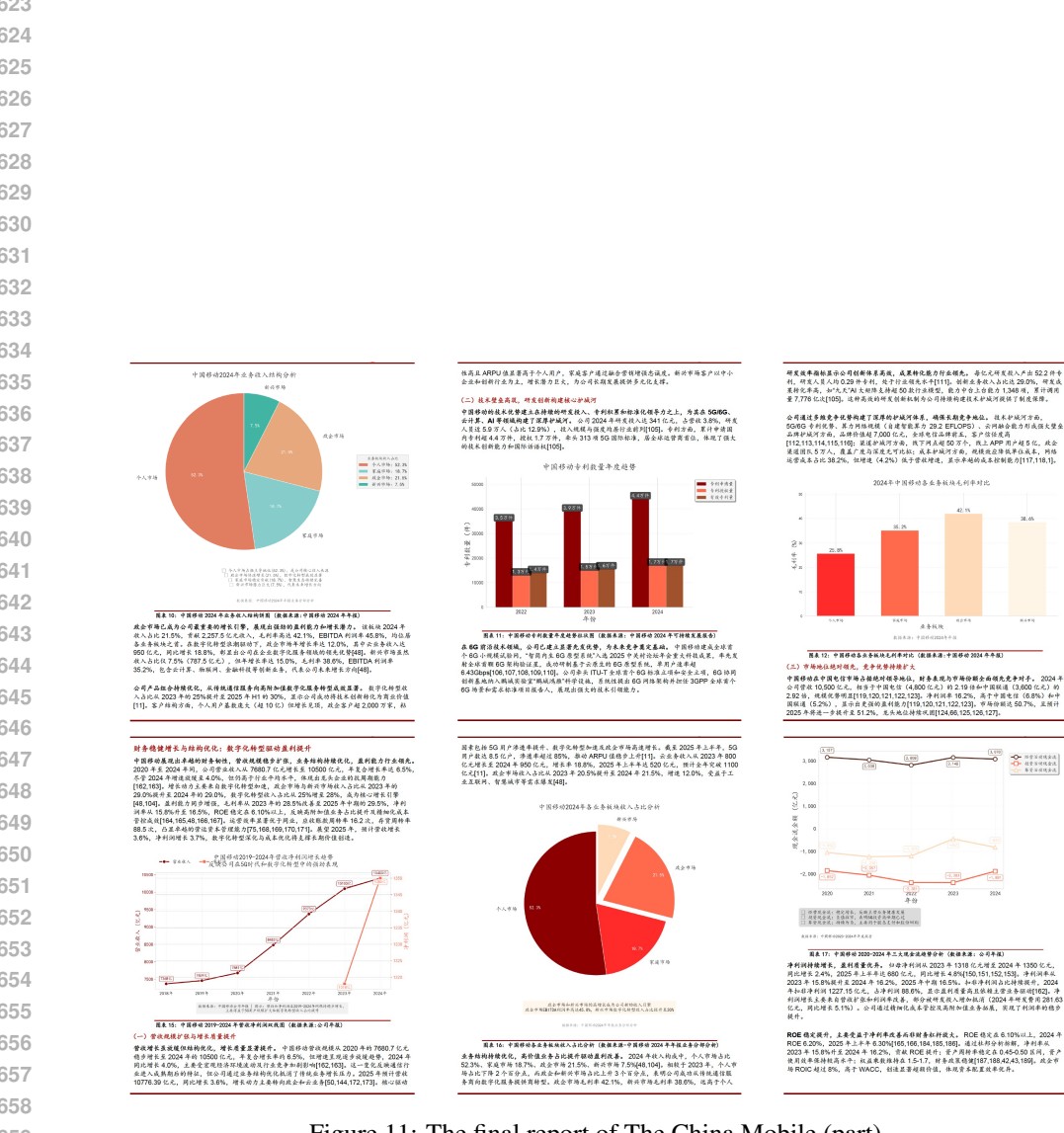

Figure 11: The final report of The China Mobile (part).

