# OpenReview forum: "FINSIGHT: TOWARDS REAL-WORLD FINANCIAL DEEP RESEARCH"
_ICLR.cc/2026/Conference — ICLR 2026 Conference Withdrawn Submission_

### Official Review · Reviewer_vZ9e · 2025-10-21

**Soundness:** 2
**Presentation:** 3
**Contribution:** 2
**Rating:** 4
**Confidence:** 4

**Summary:**

FinSight introduces a multi-agent system for automated financial report generation. Its CAVM architecture and iterative vision mechanism enable accurate data synthesis and professional-grade charts, and the two-stage writing framework enhances coherence and depth. FinSight uses LLM as judge to evaluate the qualities of generated reports, comparing with deep research models. Experimental results demonstrate that FinSight achieves the highest scores across all evaluation dimensions, outperforming both deep search agents and LLMs equipped with search tools.

**Strengths:**

1. The framework diagram of Finsight is very clear, and the overall workflow is well-structured and easy to follow.
2. The proposed two-stage writing framework substantially enhances the overall quality of report generation. Also, the iterative vision-enhanced mechanism effectively integrates textual reasoning with visual elements to meet the needs of multimodal financial analysis.

**Weaknesses:**

1. The approach largely extends the deep research agent framework to the financial domain, with improvements mainly in the writing and generation stages. While effective, the overall innovation appears somewhat limited. Also, CAVM is overclaimed as a novel agent architecture. While FinSight includes limited code-execution components for data visualization and analysis, its overall workflow is primarily coordinated through multi-agent collaboration enhanced by code execution rather than code-driven reasoning.

2. In the report generation evaluation process, Finsight uses LLM as judge to give scores on 3 different aspects. However, for the factual accuracy dimension, aspects such as citation accuracy and faithfulness are typically measurable through formula- or rule-based metrics in RAG. Relying on an LLM to subjectively score these aspects may introduce bias or inconsistency.

3. In the experiments, FinSight was compared with LLMs with search tools and deep research agents. As shown in Figure 4, the reports generated by FinSight are significantly longer and get better scores. However, this correlation between report length and quality score may not be entirely desirable. It could suggest a potential bias in the LLM-based evaluation, which might favor longer outputs while paying less attention to their actual quality. A relatively fair comparison could be achieved by controlling the report length across methods.
4. The paper does not include the implementation code, which may limit the reproducibility of the proposed method.

**Questions:**

1. How is code execution integrated into the FinSight workflow? From the overall framework, it seems that the writing stage mainly involves text generation rather than executable code.
2. Could the authors clarify whether the proposed *variable memory* primarily serves as a conceptual abstraction for flexible variable access, or if it introduces a genuinely new representation or learning mechanism in memory design?

---

> ### Author Response · Authors · 2025-11-21
> **Author Response**
>
> Dear Reviewer vZ9e,
>
> We sincerely thank the reviewer for the constructive feedback and for recognizing the clarity of our framework, the effectiveness of the Two-Stage Writing strategy, and the innovation of the Iterative Vision-Enhanced Mechanism. Below, we address your concerns regarding novelty, the definition of CAVM, and the evaluation methodology.
>
> ### Response to W1 & Q1
>
> We apologize if the distinction between our work and existing Deep Research systems was not sufficiently clear. We wish to clarify two key aspects: (1) How FinSight differs from generic Deep Research frameworks, and (2) The specific role of code execution within our workflow.
>
> ### Distinctions from Other Deep Research Systems
>
> As analyzed in our Related Work, existing Deep Research systems are primarily limited to **text-only report generation** and possess only basic **Web search** capabilities, lacking the ability to acquire, process, and deeply analyze complex data. These two limitations are the core problems this study aims to address.
>
> We wish to clarify that **this study is not merely a simple extension of existing Deep Research systems into the financial domain.** Instead, we selected "Financial Report Generation" as our experimental setting specifically because its **data-intensive** and **multimodal** characteristics serve as an ideal testbed to validate our architecture's advantages in handling complex tasks.
>
> To address the aforementioned limitations, we introduce three innovations that have not been explored in existing works:
>
> 1.  **Code Agent with Variable Memory (CAVM):** We designed the CAVM mechanism to uniformly manage complex data (especially long-context data like numeric tables and reports), tools (various data acquisition and search APIs), and intermediate variables. This mechanism empowers agents with the flexibility to collect, process, and calculate data, breaking the limitations of traditional context windows and enabling efficient data flow between different agents.
> 2.  **Iterative Vision-Enhanced Mechanism:** To address the lack of aesthetic and professional quality in chart generation found in existing works (Section 2.4), we introduced this mechanism. It leverages visual feedback to iteratively refine charts, thereby maximizing information density and achieving professional, publication-grade quality.
> 3.  **Deep Analysis Framework:** Traditional RAG or general QA-based DeepSearch methods struggle to meet the **analytical depth** required for financial reports (including the deduction of deep insights and precise calculation of intermediate data). To overcome this, we proposed the **Two-Stage Writing Framework** (Section 2.5). This approach combines generative capabilities with structured writing to seamlessly integrate citations, data calculations, and visualizations into long-form reports.
>
> ### The Role of CAVM and Code Execution
>
> Your understanding is partially correct. We wish to clarify that the **CAVM mechanism** is not designed for "code-driven reasoning" in the symbolic sense (e.g., theorem proving). Instead, it serves as a  **Context Management System** that uses programmable variables to store and invoke information. This is particularly critical for handling extensive financial data, such as long tables and multi-turn search results, which often overwhelm standard context windows. To give a more comprehensive introduction, we detail how CAVM facilitates code execution in each agent below.
>
> **Overall Mechanism:**
> Each agent possesses its own **Variable Space**. Upon initialization, relevant tools and data are injected into this space as executable objects.
>
> * **Data Collection Agent:** This agent is equipped with tools for external data acquisition. In each execution round, it autonomously plans and **invokes tools via code**. It then inspects the returned output and persists the valid results into the global variable space for downstream use.
>
> * **Data Analysis Agent:**The variable space of this agent is populated with all previously collected data, along with an interface to call the `DeepSearch Agent`. During execution, the analyzer employs code for a dual purpose:
> 1.  **Data Processing:** Retrieving data variables to perform precise calculations or generate plots.
> 2.  **Dynamic Retrieval:** Invoking the `DeepSearch Agent` via code functions to acquire supplementary information when the current data is insufficient.

---

> > ### Author Response · Authors · 2025-11-21
> >
> > * **Writing Agent (Report Generation):** Contrary to a purely generative approach, our Writing Agent also relies heavily on code execution. Its variable space includes raw data, analysis results, and the `DeepSearch` tool.
> > While it does not perform heavy calculations, it uses code for **getting existing data** and **on-demand data fethcing**(through deepsearch agent). The writing process is a multi-round loop. The agent first executes code to "read" specific variables from the memory or fetch new external data if a gap is identified. Only after gathering sufficient context does it generate the final text content. This "Code-Augmented Retrieval" strategy ensures **contextual conciseness**. Instead of stuffing all available data into the prompt, the agent actively decides **which variables to load via code**, significantly **reducing hallucination** and improving coherence.
> >
> > ### Response to Q2: The Understanding of Variable Memory
> >
> > We clarify that the **Variable Memory** in CAVM is proposed as a **novel state representation mechanism for agent**, rather than a learning mechanism in memory. It introduces a fundamental shift in how agents interact with information: shifting from **"reading context"** to **manipulating variables.**  We design this architecture specifically because our scenario necessitates interacting with extensive datasets and performing secondary processing (e.g., precise calculations and plotting) on that data. This stands in contrast to standard Deep Research agents, which are typically limited to merely summarizing text and extracting information.
> >
> > While traditional agents (e.g., ReAct) rely on unstructured text or vector embeddings as memory (which is passive and prone to hallucination in calculation), **Variable Memory** structures the environment into a **Programmable Variable Space ($\mathcal{V}$)**. This design is new in the context of Deep Research Agents architectures for two reasons:
> >
> > 1.  **Executable Representation:** Unlike text-based memory, data in our system is stored as executable Python objects (e.g., `pandas.DataFrame` for financial tables, `JSON` for competitor lists). This allows the agent to perform rigorous mathematical operations via code, which is critical for financial accuracy, rather than relying on the LLM's next-token prediction for arithmetic.
> > 2.  **Unified Manipulation:** We map heterogeneous elements—Data ($\mathcal{V}_{data}$), Tools ($\mathcal{V}_{tool}$), and Agents ($\mathcal{V}_{agent}$) into a unified variable scope. This allows an agent to call another agent as a function (e.g., `call_agent()`) or process data using the same code interface, enabling hierarchical and recursive reasoning that static context windows cannot support.
> >
> > In summary, we consider this to be the most significant distinction and advantage of our system compared to existing deep research frameworks.
> >
> >
> > ### Response to W2: Evaluation Rigor and Hallucination
> >
> > We appreciate this insightful comment, which echoes concerns raised by other reviewers. The evaluting of factuality is significantly challenging as the generation of these multimodal reports are complex. Standard metrics commonly used in RAG tasks (e.g., Accuracy, Token-Level F1) are inapplicable here due to the absence of strict ground truth. To rigorously address your concern and bridge this evaluation gap, we have conducted supplementary evaluations **using both human experts and rule-based methods** to verify factual accuracy and citation precision. Furthermore, we conducted a study where human experts replaced the LLM-as-a-Judge to score reports across all dimensions, allowing us to verify human-LLM alignment and investigate potential biases in the automated evaluation.
> >
> > #### 1. Key Fact Accuracy
> > Directly measuring the factuality of long reports is challenging. To quantify this, we introduced a **Golden Facts Evaluation**. We extracted **13 core financial indicators** (Ground Truth) from the professional Golden Reports, covering Profitability (e.g., Gross Margin), Growth (e.g., Revenue Growth), Financial Health (e.g., Cash Flow), Valuation (e.g., PE Ratio), and Efficiency (e.g., ROE). Then, we manually verified how many of these specific data points were accurately retrieved and reported by each model across company-level tasks.
> >
> > **Table 1: Key Information Recall**
> >
> > | Method | Avg. Hits (out of 13) | Avg. Recall Rate | Relative Performance |
> > | :--- | :---: | :---: | :--- |
> > | **FinSight (Ours)** | **7.1** | **54.6%** | -- |
> > | Gemini-2.5-Pro Deep Research | 5.0 | 38.5% | -29.6% |
> > | OpenAI Deep Research | 3.9 | 30.0% | -45.1% |
> >
> > Result shows that FinSight achieves a significantly higher recall rate, demonstrating superior coverage of critical financial data compared to commercial deep research systems.

---

> > > ### Author Response · Authors · 2025-11-21
> > >
> > > #### 2. Citation Accuracy
> > > We also manually verified the **Citation Faithfulness**. Experts checked the top 50 citations in generated reports to verify if the cited source actually supported the generated claim.
> > >
> > > **Table 2: Citation Verification**
> > >
> > > | Metric | FinSight (Ours) | Gemini-2.5-Pro Deep Research |
> > > | :--- | :---: | :---: |
> > > | **Total Citations Checked** | 469 | 414 |
> > > | **Overall Accuracy** | **72.92%** (342/469) | 69.81% (289/414) |
> > >
> > > Even while generating a higher volume of citations, FinSight maintains higher accuracy. We attribute this to our **Two-Stage Writing Framework** and generative retrieval mechanism, which identifies references during the drafting process rather than via post-hoc appending.
> > >
> > >
> > > #### 3. Citation Authority Analysis
> > > We further analyzed the quality of sources to determine the "Authority" of the information used. We classified the top 50 citations from each report into three categories using an LLM-based classifier (verified for accuracy):
> > > * **High Authority:** Government/Regulatory bodies (SEC, IMF), Official Company Filings, Top Academic/Research Institutions.
> > > * **Medium Authority:** Mainstream Financial Media (Bloomberg, Reuters), Known Market Research Firms.
> > > * **Low Authority:** Social Media, Personal Blogs, Content Farms, or unverified aggregators.
> > >
> > > **Table 3: Distribution of Citation Authority**
> > >
> > > | Model | High Authority | Medium Authority | Low Authority | Total |
> > > | :--- | :---: | :---: | :---: | :---: |
> > > | **Gemini-2.5-Pro Deep Research** | 312 (36.7%) | 304 (35.8%) | 233 (27.4%) | 849 |
> > > | **OpenAI Deep Research** | 348 (35.0%) | 504 (50.7%) | 142 (14.3%) | 994 |
> > > | **FinSight (Ours)** | **334 (36.5%)** | 319 (34.8%) | 263 (28.7%) | 916 |
> > >
> > > Table 3 shows that FinSight utilizes **High Authority** sources at a rate comparable to Gemini-2.5-Pro (~36.5%). We also observe a slightly higher usage of Low Authority sources compared to OpenAI DR and Gemini DR, which is maybe a limit due to our using open web search versus proprietary filtered databases. This indicates a future direction for refining our retrieval filtering modules.
> > >
> > >
> > >
> > > #### 4. Human Expert Evaluation
> > >
> > > We recruited **6 graduate students with financial backgrounds** to serve as expert annotators. To ensure robust evaluation, we selected the two strongest baselines—**Gemini-2.5-Pro Deep Research** and **OpenAI Deep Research**—to compare against **FinSight**. Each annotator reviewed a random subset of 10 research topics.
> > >
> > > **Methodology:**
> > > * **Scoring:** To manage cognitive load, raters scored on a 0–5 scale (0.5 increments) across three dimensions: *Factual*, *Analytical*, and *Presentation*. These were scaled ($\times 2$) to align with our 0–10 automated metrics.
> > > * **Reference:** Raters were provided with "Golden Reports" (professional analyst reports) as ground truth references.
> > > * **Consolidated Score:** We averaged the corresponding sub-metrics from the automated evaluation to compare against human scores.
> > >
> > > We calculated Inter-Rater Reliability (Krippendorff’s Alpha) and Human-LLM Correlation (Pearson $r$).
> > >
> > > **Table 4: Human Evaluation Scores (0-10 Scale)**
> > >
> > > | Model | Factual | Analytical | Presentation | **Total Score** |
> > > | :--- | :---: | :---: | :---: | :---: |
> > > | **OpenAI Deep Research** | 5.93 | 5.81 | 4.75 | 5.50 |
> > > | **Gemini-2.5-Pro Deep Research** | **6.86** | 6.73 | 4.73 | 6.11 |
> > > | **FinSight (Ours)** | 6.68 | **7.17** | **7.48** | **7.11** |
> > >
> > > **Table 5: Alignment Metrics**
> > >
> > > | Dimension | Human-LLM Alignment (Pearson $r$) | Human Inter-Rater (Krippendorff's $\alpha$) |
> > > | :--- | :--- | :--- |
> > > | **Factual** | 0.6360 | 0.4667 |
> > > | **Analytical** | 0.6003 | 0.4752 |
> > > | **Presentation** | 0.6757 | 0.8570 |
> > > | **Total Score** | **0.7587** | **0.6474** |
> > >
> > > From these results, we have some key observations:
> > > 1.  **Superior Performance with Justifiable Gaps:** FinSight achieves the highest **Total Score (7.11)**, significantly outperforming both commercial baselines. While Gemini-2.5-Pro holds a slight edge in the *Factual* dimension (6.86 vs. 6.68), we attribute this to its access to **proprietary, commercial-grade search tools**, whereas FinSight relies on open web search engine. However, FinSight dominates in *Analytical Depth* and *Presentation Quality*, proving the effectiveness of our specialized agentic workflow.
> > > 2.  **Validation of Automated Metrics:** The strong positive correlation between human and LLM scoring (Pearson $r > 0.75$ for Total Score) **validates the reliability of the automated evaluation framework** used in our main paper. This confirms that our LLM-based judges serve as effective proxies for human experts in assessing complex financial reports.

---

> > > > ### Author Response · Authors · 2025-11-21
> > > >
> > > > 3.  **Robust Inter-Rater Agreement:** The overall inter-rater reliability ($\alpha=0.64$) indicates a solid consensus among experts. Notably, the exceptionally high agreement in the *Presentation* dimension confirms that FinSight's multimodal capabilities (text-chart integration) provide a clear  advantage. While agreement is slightly lower for Factual/Analytical dimensions due to the inherent subjectivity in evaluating financial analysis, the scores remain within the moderate agreement range ($\alpha > 0.4$).
> > > >
> > > > ### Response to W3: Length Bias and Controlled Evaluation
> > > > We sincerely thank the reviewer for pointing out the potential bias introduced by report length in LLM-based evaluations. We acknowledge that longer reports may inherently score higher due to information abundance, and a length-controlled comparison is crucial for a fair assessment.
> > > >
> > > > To address this concern, we conducted an additional experiment where we imposed a strict length constraint on FinSight generated reports. We applied a truncation strategy to limit the output to approximately 10,000 words, aligning it more closely with the token consumption of the baseline models. The comparative results are presented in the table below:
> > > >
> > > > **Table 6: Length-Controlled Evaluation Scores**
> > > > | Method | **Factual**<br>Consis. | **Factual**<br>Faith. | **Factual**<br>T-I. | **Analytical**<br>Rich. | **Analytical**<br>Cover. | **Analytical**<br>Ins. | **Pres.**<br>Logic | **Pres.**<br>Lang. | **Pres.**<br>Vis. | **Overall** |
> > > > | :--- | :---: | :---: | :---: | :---: | :---: | :---: | :---: | :---: | :---: | :---: |
> > > > | **OpenAI Deep Research** | 5.60 | 7.45 | 4.90 | 6.35 | 6.40 | 5.90 | 6.90 | 6.85 | 4.65 | 6.11 |
> > > > | **Gemini-2.5-Pro Deep Research** | 7.10 | 6.80 | 4.65 | 7.45 | 7.75 | 7.85 | 7.65 | 7.85 | 4.25 | 6.82 |
> > > > | **Finsight** | 6.85 | 7.50 | 7.85 | 8.70 | 8.30 | 8.45 | 8.05 | 8.10 | 9.00 | **8.09** |
> > > > | **Finsight (cut off to 10k length)** |  6.20 | 6.70 | 7.00 | 6.60 | 6.75 | 7.05 | 6.60 | 7.50 | 8.05 | <u>6.93</u> |
> > > >
> > > > Even with the length constraint (which naturally penalizes coherence and completeness due to forced truncation), Finsight (10k) achieves an Overall score of 6.93, which still surpasses Gemini-2.5-Pro (6.82) and significantly outperforms OpenAI Deep Research (6.11). This demonstrates that FinSight's performance gain is derived from high-quality content synthesis rather than mere verbosity.
> > > > Metrics such as Analytical Richness and Presentation Logic saw a decline compared to the full FinSight version. This is likely because rigid truncation can disrupt the narrative flow and reduce the breadth of data coverage. However, the model's ability to maintain high scores in Analytical Insight (7.05) suggests that the core reasoning capabilities remain intact.
> > > >
> > > >
> > > > ### Response to W4: Code and Reproducibility
> > > >
> > > > We agree that reproducibility is essential. While we were unable to provide the code at the time of submission due to internal clearance processes, we have now organized and released the complete code.
> > > >
> > > > * **Code Release:** The source code, including the multi-agent design and CAVM architecture implementation, is available at this anonymous repository: **[https://anonymous.4open.science/r/FinSight-6739/](https://anonymous.4open.science/r/FinSight-6739/)**.
> > > > * **Data Release:** Due to copyright and legal restrictions regarding the proprietary financial reports used as ground truth, we cannot publicly release the dataset at this moment. We are actively working on a solution to share a subset of the data or descriptors in the future.
> > > >
> > > > We once again thank you for your thorough and valuable suggestions. If you have any questions, please feel free to raise them, and we will do our best to address them.

---

> ### Author Response · Authors · 2025-11-26
> **Revision Update**
>
> Dear Reviewer:
>
> We would like to express our gratitude for your time and constructive feedback.
>
> Following your suggestions, we have just uploaded a revised version of our manuscript. To facilitate your review, we have **highlighted all major modifications in red**.
> In response to the constructive feedback received, we have made extensive revisions to both the main text and the appendix. We hope these substantial updates effectively address your concerns. We would greatly appreciate it if you could share your thoughts on the revised version. We are happy to engage in further discussions.
>
> Thank you for your time and effort in reviewing.
>
> Best regards, The Authors

---

### Official Review · Reviewer_9Dq6 · 2025-10-24

**Soundness:** 1
**Presentation:** 2
**Contribution:** 2
**Rating:** 2
**Confidence:** 5

**Summary:**

This paper presents FinSight multi-agent system for generating long-form, multimodal financial research reports. Two main contributions are a Code Agent with Variable memory framework that unifies data, tools, and agents together with code execution, and a two stage writing framework with Chains of Analysis to produce structured reports. A vision-enhanced loop is also included to critique and refine code-generated charts (stock price charts, etc). The paper provides several examples in the appendix.

The paper outlines the main problem as producing a unified text, visual, and citation-grounded report with access to real-time data (via tools) and grounded content generation (VLM critic, graphs, visual generation). The system is evaluated on a small 20 target benchmark and evaluated on 9 metrics by an LLM-as-Judge (Gemini-2.5-pro). The framework is compared against several commercial dee research platforms (OpenIAI, Grok, Deepseek).

**Strengths:**

1. System design is coherent and includes formulations for the CAVM updates. There is a clear decomposition and separation of modalities with a programmable memory abstraction.
2. Integration of modalities is well defined and motivated, and examples in appendix look good.
3. Iteration on stock charts highlights the core contrition of the VLLM critic. This is well-motivated, and the iterative model, factor analysis is well specified.
4. Generally readable minus 1 typo.
5. Ablations cover each component and experiments cover wide variety of commercial research platforms.
6. Transparency on toolchain, judge prompts is very helpful for reproducibility.

**Weaknesses:**

1. LLM-as-judge without calibration or human study. Scores are produced by Gemini-2.5-Pro using list-wise prompts, but the paper does not report inter-rater reliability, judge sensitivity to prompt phrasing, or correlation with human experts. No robustness analysis (e.g., seed variation, alternative judges, bootstrap CIs) is provided. The bias in gemini is evident from the experiments (gemini is often 2nd best).
2. Small, in-distribution benchmark. There are only 20 examples, and the formats/layouts/color schemes are similar. There is no OOD test on alternative firm reports or other market/regulators (EU reports, EDGAR) - generalizability is unclear.
3. The decision to restrict google search api to a single region -> is this applied to all models? How are the authors mitigating regional bias from one provider vs another. This should be made clearer.
4. Metric definitions are all subjective, given the small evaluation set the anticipated variability on repeated judgements is not clear. what is the CI on these numbers? Furthermore, evaluation is qualitative - this paper would benefit from some quantitative metrics (difference in pixels from real data vs generated chart, etc.)

**Questions:**

1. What is the agreement between gemini and human labels? What is the sensitivity to prompt variants and stochastic generations? 20 samples is a small set, what are the confidence intervals and variability in measurements under repeated analysis?
2. Beyond LLM scoring, did you evaluation precision/recall on citations or fact checking? How did you evaluate that claims are grounded in sources? How are you auditing the factual and quantitative claims in the reports? Consider adding a human review study of the reports for factuality and hallucination detection.
3. Baselines: How did you ensure each baseline had the same regional access to data?
4. Can you comment on plans for out of domain generalization experiments? Have you considered public filings from other jurisdictions (US annual reports, 10Ks, EU annual reports). These are mandated to be public, which helps reproducibility.

---

> ### Author Response · Authors · 2025-11-21
> **Author Response**
>
> Dear Reviewer 9Dq6,
>
> We sincerely thank you for the detailed assessment and for recognizing the novelty of our **Code Agent with Variable Memory (CAVM)** and the **Iterative Vision-Enhanced Mechanism**.
>
> We acknowledge that as an early exploration into **domain-specific, data-intensive, and multimodal report generation**, our initial evaluation relied heavily on automated metrics. To rigorously address your concerns regarding evaluation rigor, bias, and quantitative grounding, we have conducted **extensive supplementary experiments**, including a human expert study, cross-model validation, and granular quantitative auditing of facts and citations. We will incorporate all these revisions into the final version of the paper.
>
> ### Response to W1 & Q1:
>
> We sincerely appreciate the reviewer for highlighting the limitations of relying solely on a single LLM-as-Judge. To rigorously address your concerns and validate our claims, we conducted a comprehensive **three-pronged supplementary evaluation**:
> 1.  **Stability Analysis:** We performed repeated runs (3x) with Gemini-2.5-Pro to calculate Confidence Intervals (CI) and assess scoring consistency.
> 2.  **Bias Mitigation:** We employed **GPT-5** as an alternative evaluator to determine if the original scores were biased by the judge model's self-preference.
> 3.  **Human Evaluation:** We recruited human experts to assess the reports and calculated the correlation between human and LLM scores to validate the automated judge.
>
> #### 1. Stability of Automated Metrics
> To measure the sensitivity of our judge to stochastic variations, we repeated the evaluation process three times. The 95% Confidence Intervals (CI) and standard deviations are reported in **Table 1**.
>
> **Table 1: 95% Confidence Intervals across 3 evaluation runs.**
> | Method | **Factual**<br>Consis. | **Factual**<br>Faith. | **Factual**<br>T-I. | **Analytical**<br>Rich. | **Analytical**<br>Cover. | **Analytical**<br>Ins. | **Pres.**<br>Logic | **Pres.**<br>Lang. | **Pres.**<br>Vis. | **Overall** |
> | :--- | :---: | :---: | :---: | :---: | :---: | :---: | :---: | :---: | :---: | :---: |
> | **GPT5 w/ Search** | 5.95 ± 0.48 | 6.35 ± 0.39 | 4.77 ± 0.72 | 5.43 ± 0.58 | 4.52 ± 0.51 | 5.09 ± 0.49 | 6.53 ± 0.27 | 5.87 ± 0.35 | 3.90 ± 0.79 | 5.38 ± 0.51 |
> | **Claude-4.1-Sonnet w/ Search** | 5.78 ± 0.52 | 5.92 ± 0.55 | 3.55 ± 0.65 | 5.58 ± 0.49 | 5.25 ± 0.51 | 5.01 ± 0.46 | 6.34 ± 0.26 | 6.07 ± 0.40 | 2.59 ± 0.51 | 5.12 ± 0.48 |
> | **DeepSeek-R1 w/ Search** | 6.26 ± 0.36 | 5.92 ± 0.44 | 4.08 ± 0.37 | 6.68 ± 0.26 | 6.33 ± 0.34 | 6.62 ± 0.41 | 7.03 ± 0.20 | 6.79 ± 0.29 | 3.35 ± 0.45 | 5.90 ± 0.35 |
> | **Grok Deep Search** | 4.71 ± 0.81 | 5.72 ± 0.59 | 4.21 ± 0.57 | 4.90 ± 0.55 | 4.03 ± 0.58 | 4.35 ± 0.55 | 5.87 ± 0.42 | 5.61 ± 0.48 | 3.76 ± 0.53 | 4.79 ± 0.56 |
> | **Perplexity Deep Research** | 5.02 ± 0.61 | 5.74 ± 0.63 | 4.03 ± 0.97 | 3.88 ± 0.60 | 3.40 ± 0.53 | 3.65 ± 0.59 | 5.47 ± 0.44 | 4.92 ± 0.47 | 3.42 ± 0.97 | 4.39 ± 0.65 |
> | **Gemini-2.5-Pro Deep Research** | 5.92 ± 0.61 | 6.66 ± 0.56 | 4.32 ± 0.91 | 6.19 ± 0.65 | 6.03 ± 0.61 | 5.74 ± 0.62 | 6.77 ± 0.40 | 6.70 ± 0.43 | 3.23 ± 0.95 | 5.73 ± 0.64 |
> | **OpenAI Deep Research** | **6.87 ± 0.52** | 6.78 ± 0.34 | 4.58 ± 0.51 | 6.79 ± 0.35 | 6.83 ± 0.30 | 7.33 ± 0.40 | 7.56 ± 0.25 | 7.58 ± 0.31 | 3.66 ± 0.50 | 6.44 ± 0.39 |
> | **Finsight** | 6.84 ± 0.59 | **7.59 ± 0.52** | **7.84 ± 0.52** | **8.49 ± 0.47** | **8.44 ± 0.50** | **7.78 ± 0.48** | **7.82 ± 0.38** | **7.98 ± 0.33** | **8.57 ± 0.57** | **7.93 ± 0.49** |
>
> **Table 2: Standard deviation across specific dimensions.**
> | Dimension | Avg Std. |
> | :--- | :--- |
> | Core Conclusion Consistency (Cons.) | $1.010$ |
> | Textual Faithfulness (Faith.) | $1.213$ |
> | Text-Image Coherence (T-I.) | $1.470$ |
> | Information Richness (Rich.) | $1.058$ |
> | Information Coverage (Cover.) | $0.966$ |
> | Analytical Insight (Ins.) | $0.932$ |
> | Structural Logic (Logic) | $0.675$ |
> | Professional Language (Lang.) | $0.848$ |
> | Chart Expressiveness (Vis.) | $1.340$ |
>
>
> Based on Table 1 and Table 2, we have two key observations:
> * **Low Variance:** As shown in the table, the standard deviations for the 95% confidence intervals are consistently within **1.0 point** across all metrics. We attribute this stability to our robust evaluation protocol, which anchors the scoring process against a provided "Golden Report" and utilizes a detailed, list-wise grading rubric. This context prevents the model from drifting significantly between runs.
> * **Dimension-Specific Volatility:** We analyzed the standard deviation across specific dimensions. While structural and linguistic metrics (e.g., *Structural Logic*, *Analytical Insight*) showed high stability (Std Dev < 1.0), we observed slightly higher variance in **Factual metrics** (e.g., *Textual Faithfulness*). This indicates that LLM judges are inherently more sensitive when verifying facts compared to assessing writing style. However, the overall ranking trend remained unaffected.

---

> > ### Author Response · Authors · 2025-11-21
> >
> > #### 2. Cross-Model Verification
> > To investigate whether Gemini-2.5-Pro exhibited "self-preference bias" (favoring its own outputs), we introduced **GPT-5** as an independent evaluator using the exact same prompt and rubric. The table 3 below contains the ratings and ranks of two evaluation models for different baselines.
> >
> > **Table 3: Comparison of Evaluator Scores**
> > | Method | Avg Score (GPT-5) | Avg Score (Gemini-2.5 Pro) | Rank (GPT-5) | Rank (Gemini-2.5 Pro) |
> > |:---:|:---:|:---:|:---:|:---:|
> > | GPT-5 w/ Search | 6.63 | 5.38 | 4.0 | 5.0 |
> > | Claude-4.1-Sonnet w/ Search | 4.71 | 5.12 | 7.0 | 6.0 |
> > | DeepSeek-R1 w/ Search | 6.75 | 5.90 | 3.0 | 3.0 |
> > | Grok Deep Search | 4.71 | 4.79 | 8.0 | 7.0 |
> > | Perplexity Deep Research | 6.08 | 4.39 | 6.0 | 8.0 |
> > | OpenAI Deep Research | 6.19 | 5.73 | 5.0 | 4.0 |
> > | Gemini-2.5-Pro Deep Research | 6.85 | 6.44 | 2.0 | 2.0 |
> > | Finsight (ours) | 8.04 | 7.93 | 1.0 | 1.0 |
> >
> > The ranking order remained highly consistent between the two judges (Kendall’s $\tau = 0.764$, $p=0.008$). Notably, even when evaluated by GPT-5, **Gemini-2.5-Pro DR retained the second-place position**, and **FinSight retained the top spot**. This confirms that FinSight's high performance is due to objective report quality rather than evaluator bias.
> >
> > #### 3. Human Expert Evaluation
> >
> > We recruited **6 graduate students with financial backgrounds** to serve as expert annotators. To ensure robust evaluation, we selected the two strongest baselines—**Gemini-2.5-Pro Deep Research** and **OpenAI Deep Research**—to compare against **FinSight**. Each annotator reviewed a random subset of 10 research topics.
> >
> > **Methodology:**
> > * **Scoring:** To manage cognitive load, raters scored on a 0–5 scale (0.5 increments) across three dimensions: *Factual*, *Analytical*, and *Presentation*. These were scaled ($\times 2$) to align with our 0–10 automated metrics.
> > * **Reference:** Raters were provided with "Golden Reports" (professional analyst reports) as ground truth references.
> > * **Consolidated Score:** We averaged the corresponding sub-metrics from the automated evaluation to compare against human scores.
> >
> > We calculated Inter-Rater Reliability (Krippendorff’s Alpha) and Human-LLM Correlation (Pearson $r$).
> >
> > **Table 4: Human Evaluation Scores (0-10 Scale)**
> >
> > | Model | Factual | Analytical | Presentation | **Total Score** |
> > | :--- | :---: | :---: | :---: | :---: |
> > | **OpenAI Deep Research** | 5.93 | 5.81 | 4.75 | 5.50 |
> > | **Gemini-2.5-Pro Deep Research** | **6.86** | 6.73 | 4.73 | 6.11 |
> > | **FinSight (Ours)** | 6.68 | **7.17** | **7.48** | **7.11** |
> >
> > **Table 5: Alignment Metrics**
> >
> > | Dimension | Human-LLM Alignment (Pearson $r$) | Human Inter-Rater (Krippendorff's $\alpha$) |
> > | :--- | :--- | :--- |
> > | **Factual** | 0.6360 | 0.4667 |
> > | **Analytical** | 0.6003 | 0.4752 |
> > | **Presentation** | 0.6757 | 0.8570 |
> > | **Total Score** | **0.7587** | **0.6474** |
> >
> > From these results, we have some key observations:
> > 1.  **Superior Performance with Justifiable Gaps:** FinSight achieves the highest **Total Score (7.11)**, significantly outperforming both commercial baselines. While Gemini-2.5-Pro holds a slight edge in the *Factual* dimension (6.86 vs. 6.68), we attribute this to its access to **proprietary, commercial-grade search tools**, whereas FinSight relies on open web search engine. However, FinSight dominates in *Analytical Depth* and *Presentation Quality*, proving the effectiveness of our specialized agentic workflow.
> > 2.  **Validation of Automated Metrics:** The strong positive correlation between human and LLM scoring (Pearson $r > 0.75$ for Total Score) **validates the reliability of the automated evaluation framework** used in our main paper. This confirms that our LLM-based judges serve as effective proxies for human experts in assessing complex financial reports.
> > 3.  **Robust Inter-Rater Agreement:** The overall inter-rater reliability ($\alpha=0.64$) indicates a solid consensus among experts. Notably, the exceptionally high agreement in the *Presentation* dimension confirms that FinSight's multimodal capabilities (text-chart integration) provide a **clear, objectively recognizable advantage**. While agreement is slightly lower for Factual/Analytical dimensions due to the inherent subjectivity in evaluating financial analysis, the scores remain within the moderate agreement range ($\alpha > 0.4$), ensuring the validity of the results.

---

> > > ### Author Response · Authors · 2025-11-21
> > >
> > > ### Response to W2 & Q4
> > > We thank the reviewer for the thoughtful questions regarding our evaluation setup and generalization capabilities. We address these concerns as follows:
> > >
> > > #### 1. Benchmark Construction and Diversity
> > > As this is a pioneering work in multimodal, long-form financial report generation, no existing benchmarks were suitable for this task. Following the scale and methodology of related deep research works and survey generation works[1, 2], we constructed a high-quality, expert-aligned benchmark. We have the following two considerations
> > >
> > > **Diversity over Quantity**: While the dataset contains 20 samples, it was carefully curated to ensure **high diversity and difficulty**. It covers two distinct types of targets (specific companies vs. entire industries), spans 10 different domain sectors, and involves two distinct financial markets (A-shares and HK-stocks). This complexity sufficiently challenges the agent's ability to handle varied data structures and analytical logic.
> > >
> > > **Scope of Generalization**: Our current focus is on "Financial Deep Research." We define generalization within this context as the ability to analyze a wide variety of companies and industries within the financial domain. While EU reports or US 10-Ks are valuable data sources, extending to non-Asian markets represents a geographical expansion rather than a fundamental methodological gap. Our current benchmark effectively tests the system's core capabilities in data reasoning and report composition.
> > >
> > > #### 2. Clarification on Format and Visual Similarity
> > > You noted that the formats and color schemes are similar across reports. We wish to clarify that this is a deliberate design choice, not a limitation. In real-world financial analysis, reports from top brokerages adhere to strict formatting guidelines to ensure readability and professional consistency. Similarly, standardizing the output style (e.g., using a consistent color palette for charts) allows the evaluation to focus on the content quality (accuracy, depth, insight) rather than stylistic variance.
> > >
> > >
> > > #### 3. Future Plan in Domain Generalization
> > > We appreciate this insightful comment for domain generalization of our framework, it is indeed part of our future roadmap. As described in the paper, the overall design of our framework is domain-agnostic (particularly the CAVM architecture and the Two-Stage Writing Framework). It is fundamentally applicable to any scenario that requires aggregating data, utilizing tools, and generating figures to produce reports. The current implementation is tailored to the financial domain simply because the tools, data sources, and prompts have been optimized for this specific context.
> > >
> > > Our future plan is to extend this framework by introducing user-defined prompts, toolkits, and writing outlines, enabling it to generate reports for a wide variety of domains beyond finance. We believe this approach holds significant potential. However, the current paper focuses on validating the methodology within the financial domain; expanding to other regions or tools is primarily a matter of engineering implementation.
> > >
> > >
> > > [1] LLMxMapReduce-V2: Entropy-Driven Convolutional Test-Time Scaling for Generating Long-Form Articles from Extremely Long Resources
> > > [2] WebThinker: Empowering Large Reasoning Models with Deep Research Capability

---

> > > > ### Author Response · Authors · 2025-11-21
> > > >
> > > > ### Response to W3 & Q3: Search Region Settings & Fairness
> > > >
> > > > We appreciate your scrutiny regarding the search region settings, as it is an important part in deep research system. We wish to clarify our experimental setup and justify the fairness of the comparison:
> > > >
> > > > **Region Settings:**
> > > > It is important to note that all baselines were evaluated via their official web interfaces or APIs. As commercial "black-box" systems, we have no control over their internal search routing or region settings. For our method, we set the Google Search API region to "China" specifically to align with the benchmark dataset, which consists of A-share and HK-stock companies.
> > > >
> > > > **Discussion of Fairness**
> > > > To ensure a fair comparison, we utilized English prompts for task instructions across all models, while keeping the specific entity names (e.g., company names) in their original Chinese characters. This ensures that the query intent is correctly understood by all systems. We believe this setting is fair and potentially disadvantageous to FinSight for two reasons:
> > > >
> > > > 1.  **Relevance over Bias:** In our preliminary experiments, we observed that retrieval quality is driven primarily by the language and specificity of the query rather than the region setting. Setting the region to "China" was necessary to retrieve specific local filings for the target companies.
> > > > 2.  **Commercial Advantage:** Commercial deep research systems (e.g. Gemini DR) often have access to high-quality search resources and curated financial databases. Also, they always employ sophisticated internal query rewriting that mitigates regional restrictions. In contrast, FinSight relies on the open search engine  via a standard Google Search API. Therefore, any bias in this setup likely favors the commercial baselines, making FinSight’s performance even more notable.
> > > >
> > > > ### Response to W4 & Q2: Quantitative Metrics & Human Evaluation
> > > >
> > > > We agree that supplementing LLM-based evaluations with quantitative and human-verified metrics significantly strengthens the paper. In response to your suggestion, we conducted extensive additional evaluations focusing on **Factual Accuracy**, **Citation Quality**, and **Visual Aesthetics**.
> > > >
> > > > #### 1. Key Fact Accuracy
> > > > Directly measuring the factuality of long reports is challenging. To quantify this, we introduced a **Golden Facts Evaluation**. We extracted **13 core financial indicators** (Ground Truth) from the professional Golden Reports, covering Profitability (e.g., Gross Margin), Growth (e.g., Revenue Growth), Financial Health (e.g., Cash Flow), Valuation (e.g., PE Ratio), and Efficiency (e.g., ROE). Then, we manually verified how many of these specific data points were accurately retrieved and reported by each model across company-level tasks.
> > > >
> > > > **Table 6: Key Information Recall**
> > > >
> > > > | Method | Avg. Hits (out of 13) | Avg. Recall Rate | Relative Performance |
> > > > | :--- | :---: | :---: | :--- |
> > > > | **FinSight (Ours)** | **7.1** | **54.6%** | -- |
> > > > | Gemini-2.5-Pro Deep Research | 5.0 | 38.5% | -29.6% |
> > > > | OpenAI Deep Research | 3.9 | 30.0% | -45.1% |
> > > >
> > > > Result shows that FinSight achieves a significantly higher recall rate, demonstrating superior coverage of critical financial data compared to commercial deep research systems.
> > > >
> > > > #### 2. Citation Accuracy
> > > > We also manually verified the **Citation Faithfulness**. Experts checked the top 50 citations in generated reports to verify if the cited source actually supported the generated claim.
> > > >
> > > > **Table 7: Citation Verification**
> > > >
> > > > | Metric | FinSight (Ours) | Gemini-2.5-Pro Deep Research |
> > > > | :--- | :---: | :---: |
> > > > | **Total Citations Checked** | 469 | 414 |
> > > > | **Overall Accuracy** | **72.92%** (342/469) | 69.81% (289/414) |
> > > >
> > > > Even while generating a higher volume of citations, FinSight maintains higher accuracy. We attribute this to our **Two-Stage Writing Framework** and generative retrieval mechanism, which identifies references during the drafting process rather than via post-hoc appending.

---

> > > > > ### Author Response · Authors · 2025-11-21
> > > > >
> > > > > #### 3. Citation Authority Analysis
> > > > > We further analyzed the quality of sources to determine the "Authority" of the information used. We classified the top 50 citations from each report into three categories using an LLM-based classifier (verified for accuracy):
> > > > > * **High Authority:** Government/Regulatory bodies (SEC, IMF), Official Company Filings, Top Academic/Research Institutions.
> > > > > * **Medium Authority:** Mainstream Financial Media (Bloomberg, Reuters), Known Market Research Firms.
> > > > > * **Low Authority:** Social Media, Personal Blogs, Content Farms, or unverified aggregators.
> > > > >
> > > > > **Table 8: Distribution of Citation Authority**
> > > > >
> > > > > | Model | High Authority | Medium Authority | Low Authority | Total |
> > > > > | :--- | :---: | :---: | :---: | :---: |
> > > > > | **Gemini-2.5-Pro Deep Research** | 312 (36.7%) | 304 (35.8%) | 233 (27.4%) | 849 |
> > > > > | **OpenAI Deep Research** | 348 (35.0%) | 504 (50.7%) | 142 (14.3%) | 994 |
> > > > > | **FinSight (Ours)** | **334 (36.5%)** | 319 (34.8%) | 263 (28.7%) | 916 |
> > > > >
> > > > > Table 8 shows that FinSight utilizes **High Authority** sources at a rate comparable to Gemini-2.5-Pro (~36.5%). We also observe a slightly higher usage of Low Authority sources compared to OpenAI DR and Gemini DR, which is maybe a limit due to our using open web search versus proprietary filtered databases. This indicates a future direction for refining our retrieval filtering modules.
> > > > >
> > > > > #### 4. Quantitative Image Quality Assessment
> > > > > We appreciate the suggestion to use quantitative metrics for the generated charts. We respectfully argue that pixel-wise metrics (e.g., MSE) are ill-suited here, as different rendering engines (e.g., Matplotlib) produce large pixel discrepancies even when plotting these identical data.
> > > > >
> > > > > Instead, to rigorously quantify **presentation quality** and the impact of our **Iterative Vision-Enhanced Mechanism**, we implemented three reference-free Image Quality Assessment (IQA) metrics[4]:
> > > > > 1.  **Colorfulness:** Measures the chromatic distinction between elements.
> > > > > 2.  **RMS Contrast:** Measures luminance contrast, correlating with the legibility of labels and grid lines.
> > > > > 3.  **Edge Density:** Measures information density vs. visual clutter using Canny edge detection.
> > > > >
> > > > > **Table 9: Visual Quality Ablation**
> > > > >
> > > > > | Method | Colorfulness | Contrast | Edge Density |
> > > > > | :--- | :--- | :--- | :--- |
> > > > > | **FinSight (Full)** | **32.35** | **31.71** | **0.0056** |
> > > > > | *w/o Iterative Vision Mechanism* | 15.81 | 15.47 | 0.0027 |
> > > > >
> > > > > The quantitative metrics confirm that our Iterative Vision-Enhanced Mechanism doubles the scores across all three dimensions, objectively validating that the VLM critic loop significantly improves the aesthetic quality and information density of the charts. Meanwhile, this conclusion is consistent with the high scores previously given by humans and LLM in presentations.
> > > > >
> > > > > [4] Measuring colorfulness in natural images
> > > > >
> > > > > ### Response for Reproducibility
> > > > > We agree that reproducibility is essential. While we were unable to provide the code at the time of submission due to internal clearance processes, we have now organized and released the complete code.
> > > > >
> > > > > The source code, including the multi-agent design and CAVM architecture implementation, is available at this anonymous repository: **[https://anonymous.4open.science/r/FinSight-6739/](https://anonymous.4open.science/r/FinSight-6739/)**.
> > > > >
> > > > > We once again thank you for your thorough and valuable suggestions. If you have any questions, please feel free to raise them, and we will do our best to address them.

---

> ### Author Response · Authors · 2025-11-26
> **Revision Update**
>
> Dear Reviewer:
>
> We would like to express our gratitude for your time and constructive feedback.
>
> Following your suggestions, we have just uploaded a revised version of our manuscript. To facilitate your review, we have **highlighted all major modifications in red**.
> In response to the constructive feedback received, we have made extensive revisions to both the main text and the appendix. We hope these substantial updates effectively address your concerns. We would greatly appreciate it if you could share your thoughts on the revised version. We are happy to engage in further discussions.
>
> Thank you for your time and effort in reviewing.
>
> Best regards, The Authors

---

### Official Review · Reviewer_2oEk · 2025-10-31

**Soundness:** 3
**Presentation:** 2
**Contribution:** 3
**Rating:** 4
**Confidence:** 4

**Summary:**

The paper presents FinSight, a multi-agent system for automatically generating multimodal financial research reports that integrate text, visualizations, and citations. The system addresses three challenges—domain knowledge, multimodal generation, and analytical depth—by introducing a Code Agent with Variable Memory (CAVM) for tool-based data processing, and a Two-Stage Writing Framework that transforms chains of analysis into structured reports. An Iterative Vision-Enhanced Mechanism refines visual outputs into professional-looking charts. Experiments on company- and industry-level tasks show FinSight outperforming commercial deep research systems in factual accuracy, analytical depth, and presentation quality based on automatic and LLM-based evaluations. However, no human evaluation or code release is provided. While the work tackles a novel and practically interesting problem, the paper’s structure and methodological clarity are limited, and the evaluation lacks rigor.

**Strengths:**

The topic is timely and practically relevant, addressing the automation of multimodal financial analysis—a domain rarely explored in LLM research. The proposed FinSight system integrates multiple functional agents (retrieval, visualization, and report writing) into a coherent pipeline. The idea of separating analytical reasoning from structured writing via the Two-Stage Writing Framework is conceptually sound and could be useful for other applied domains. The paper provides qualitative examples demonstrating real-world potential and shows moderate engineering effort to bridge LLM reasoning with data-driven financial reporting.

**Weaknesses:**

1. The problem formulation lacks rigor. The definition of the financial report R as a set ignores its hierarchical and sequential structure, and the mapping from query q to report R is not well formalized.
2. The paper’s overall organization is unclear, with descriptions of system components, data collection, and evaluation interleaved, making it difficult to follow the main contributions.
3. No code, data, or benchmark release, which limits reproducibility. Given the complexity of the multi-agent design, partial release or detailed pseudo-code would be essential for validation.
4. Evaluation relies entirely on automatic or LLM-based metrics, with no human expert assessment. The claim that FinSight “approaches human-expert quality” is not substantiated.

**Questions:**

1. Please clarify how “human-expert quality” was defined and whether any human evaluation was performed.
2. Do you plan to release code or data to support reproducibility?
3. How is the “financial research report” defined—does it target professional investment research or general analytical summaries?

---

> ### Author Response · Authors · 2025-11-21
> **Author Response**
>
> Dear Reviewer 2oEk,
>
> We sincerely appreciate the time and effort you dedicated to providing constructive feedback on our manuscript. We are encouraged that you recognize the timeliness and practical relevance of our topic, as well as the engineering efforts behind the FinSight system.
>
> We have taken your concerns regarding problem formulation rigor, paper structure, and evaluation very seriously. Below, we provide detailed responses and new experimental evidence (including code release and human evaluation) to address your comments. We will incorporate all these revisions into the final version of the paper.
>
>
> ### Response to W1: Problem Formulation Rigor
> Thank you for your insightful feedback. We apologize that our initial notation caused confusion regarding the structure of the report.
>
> In our paper, we intended to model the report $R$ as a sequential structure rather than a disordered set. The notation $\{r_1, r_2, ..., r_L\}$ was meant to represent a sequence where textual, visual, and citation components are interleaved.
>
> **Revised Definition:** To address your suggestion and improve rigor, we will update the definition to explicitly reflect the hierarchical and sequential nature of the report: Let $R$ be an ordered sequence of sections: $R = {S_1, S_2, ..., S_N}$, where $N$ is the total number of sections derived from the report outline $\mathcal{O}$. Furthermore, each section $S_i$ is defined as an ordered sequence of multimodal elements $r_{i,j} \in \{T, V, C\}$, representing text, visualizations, and citations, respectively.
>
> **Formalized Generation Process ($q \to R$):** We will also add the following formalization of the mapping from the research query $q$ to the report $R$, which corresponds to our Two-Stage Writing Framework (Section 2.5):
>
> * **Stage 1: Chain-of-Analysis (CoA) Generation**
>     The system generates a set of concise, multimodal analysis chains $\mathcal{A}$ based on the query $q$ and variable space $\mathcal{V}$, ensuring coverage of multiple analytical perspectives $\mathcal{P}$:
>     $$P(\mathcal{A}|q,\mathcal{V}) = P(\mathcal{P}|q,\mathcal{V}) \cdot \prod_{i=1}^{|\mathcal{P}|} P(a_{i}|p_{i}, \mathcal{V})$$
>
> * **Stage 2: Structured Report Writing**
>     The final report is generated sequentially based on an outline $\mathcal{O}$. The probability of generating $R$ is the product of generating the outline and then generating each section $S_i$, utilizing dynamically retrieved CoA segments ($\mathcal{A}_{selected}$) and relevant variables ($\mathcal{V}_{selected}$):
>     $$P(R| \mathcal{A},\mathcal{V},q) = P(\mathcal{O}|\mathcal{A},q) \cdot \prod_{i=1}^{|\mathcal{O}|} P(S_{i}|\mathcal{A}_{selected}, \mathcal{V}_{selected})$$
>
> This formulation explicitly models the "analyze-then-write" strategy and dynamic retrieval, resolving the structural ambiguity.
>
>
> ### Response to W3 & Q2: Code and Reproducibility
>
> We agree that reproducibility is essential. While we were unable to provide the code at the time of submission due to internal clearance processes, we have now organized and released the complete code.
>
> * **Code Release:** The source code, including the multi-agent design and CAVM architecture implementation, is available at this anonymous repository: **[https://anonymous.4open.science/r/FinSight-6739/](https://anonymous.4open.science/r/FinSight-6739/)**.
> * **Data Release:** Due to copyright and legal restrictions regarding the proprietary financial reports used as ground truth, we cannot publicly release the dataset at this moment. We are actively working on a solution to share a subset of the data or descriptors in the future.

---

> > ### Author Response · Authors · 2025-11-21
> >
> > ### Response to W2: Paper Organization and Contributions
> >
> > We sincerely apologize for the confusion caused by the clarity of our writing. Our original intention was to structure the manuscript into two distinct parts: the system's operational workflow (Section 2) and the evaluation methodology (Section 3).
> >
> > The structure of **Section 2 (Method)** was designed to sequentially present FinSight's novel methodology, moving from the **Overall Framework (Section 2.2)** to its three core design pillars:
> > 1.  The architectural foundation (**CAVM**, Section 2.3).
> > 2.  The core visualization mechanism (**Iterative Vision-Enhanced**, Section 2.4).
> > 3.  The report generation framework (**Two-Stage Writing**, Section 2.5).
> >
> > The references to Data Collection in Section 2.2 were merely intended to situate the subsequent analysis and writing phases within the complete agentic workflow, rather than to detail the experimental setup.
> >
> > **Revisions to Improve Organization:**
> > To address your valuable feedback and better highlight our contributions, we will make the following changes:
> > * Condense Section 2.2: We will significantly shorten the operational descriptions of Data Collection in the main text.
> > * Move Details to Appendix: All specific implementation details (e.g., Python packages, Search API settings) will be fully moved to Appendix C (Implementation Details).
> >
> > **Clarification of Core Contributions:**
> > This reorganization ensures that Section 2 focuses purely on the conceptual design of our system. Our methodology is specifically designed to address the critical challenges of generating **data-intensive**, **interleaved text-and-image** financial reports, which still remains underexplored compared to general plain-text generation:
> > 1.  **Agent Architecture for Data Intensity:** To handle comprehensive and deep analysis, we designed the **CAVM multi-agent architecture** (Section 2.3), which significantly enhances flexibility and the management of diverse data, tools, and intermediate variables compared to standard agents.
> > 2.  **Professional Visualization:** To address the lack of aesthetic and professional charts in existing works, we introduced the **Iterative Vision-Enhanced Mechanism** (Section 2.4), which iteratively refines charts to maximize information density and professional quality.
> > 3.  **Deep Analysis Framework:** To address the limitations of traditional RAG and QA-based DeepSearch in coherent long-text generation and interleaved multimodal citations, we proposed the Two-Stage Writing Framework (Section 2.5).  This approach combines generative capabilities with structured writing to seamlessly integrate citations and visualizations into long-form reports.
> >
> > ### Response to Q1: Clarification of "Human-Expert Quality"
> > In our study, "human-expert quality" is quantitatively grounded in the Golden Reference Reports described in Section 3.1. These references are professional equity research reports authored by analysts from top-tier securities firms (e.g., CITIC, Huatai), characterized by their substantial depth (avg. >20 pages), data richness, and rigorous logic.
> > In our automated evaluation setup (as shown in Appendix E), the scoring system is calibrated such that the quality of these Golden Reports serves as the benchmark anchor. Therefore, our claim of "approaching human-expert quality" was originally based on FinSight **achieving automated evaluation scores comparable to these professional benchmarks**.
> >
> > To rigorously validate this claim and address the potential leniency bias inherent in LLM-as-a-Judge metrics, we conducted the supplementary human evaluation detailed in our Response to Weakness 4, and the results from the human evaluation provide a more nuanced perspective:
> >
> > * Relative Superiority: FinSight consistently achieves the highest ratings among all AI systems, significantly outperforming commercial baselines (OpenAI/Gemini Deep Research) in Factual, Analytical, and Presentation dimensions.
> > * The Gap with Experts: While automated metrics suggested near-parity, human judges were more discerning. Using the Golden Reports as a strict standard, we observe that while FinSight significantly bridges the gap between AI and professional analysts, a distinction remains—particularly in complex analytical reasoning.
> >
> > Therefore, we refine our claim to state that FinSight establishes a new state-of-the-art for automated financial report generation that is closer to human-expert quality than existing solutions, while acknowledging that fully matching top-tier human professionals remains an ongoing challenge for the field.

---

> > > ### Author Response · Authors · 2025-11-21
> > >
> > > ### Response to W4: Human Evaluation
> > >
> > > We sincerely appreciate the reviewer for emphasizing the necessity of human assessment. We acknowledge that as an early exploration into **domain-specific, data-intensive, and multimodal long-report generation**, our initial evaluation relied heavily on automated metrics due to the high difficulty of evaluating long-form financial content.
> > >
> > > To rigorously address your concern and validate our claims, we conducted a comprehensive **Supplementary Human Evaluation Study**. Unlike standard RAG or QA tasks, financial report generation requires assessing complex reasoning and structural coherence. Therefore, our study focuses on three dimensions: (1) **Direct Human Review**, (2) **Report Accuracy (Key Fact Recall)**, and (3) **Citation Faithfulness**.
> > >
> > > We recruited **6 graduate students with financial backgrounds** to serve as expert annotators. To ensure robust evaluation, we selected the two strongest baselines—**Gemini-2.5-Pro Deep Research** and **OpenAI Deep Research**—to compare against **FinSight**. Each annotator reviewed a random subset of 10 research topics.
> > >
> > > **Methodology:**
> > > * **Scoring:** To manage cognitive load, raters scored on a 0–5 scale (0.5 increments) across three dimensions: *Factual*, *Analytical*, and *Presentation*. These were scaled ($\times 2$) to align with our 0–10 automated metrics.
> > > * **Reference:** Raters were provided with "Golden Reports" (professional analyst reports) as ground truth references.
> > > * **Consolidated Score:** We averaged the corresponding sub-metrics from the automated evaluation to compare against human scores.
> > >
> > > We calculated Inter-Rater Reliability (Krippendorff’s Alpha) and Human-LLM Correlation (Pearson $r$).
> > >
> > > **Table A: Human Evaluation Scores (0-10 Scale)**
> > >
> > > | Model | Factual | Analytical | Presentation | **Total Score** |
> > > | :--- | :---: | :---: | :---: | :---: |
> > > | **OpenAI Deep Research** | 5.93 | 5.81 | 4.75 | 5.50 |
> > > | **Gemini-2.5-Pro Deep Research** | **6.86** | 6.73 | 4.73 | 6.11 |
> > > | **FinSight (Ours)** | 6.68 | **7.17** | **7.48** | **7.11** |
> > >
> > > **Table B: Alignment Metrics**
> > >
> > > | Dimension | Human-LLM Alignment (Pearson $r$) | Human Inter-Rater (Krippendorff's $\alpha$) |
> > > | :--- | :--- | :--- |
> > > | **Factual** | 0.6360 | 0.4667 |
> > > | **Analytical** | 0.6003 | 0.4752 |
> > > | **Presentation** | 0.6757 | 0.8570 |
> > > | **Total Score** | **0.7587** | **0.6474** |
> > >
> > > From these results, we have some key observations:
> > > 1.  **Superior Performance with Justifiable Gaps:** FinSight achieves the highest **Total Score (7.11)**, significantly outperforming both commercial baselines. While Gemini-2.5-Pro holds a slight edge in the *Factual* dimension (6.86 vs. 6.68), we attribute this to its access to **proprietary, commercial-grade search tools**, whereas FinSight relies on open web search engine. However, FinSight dominates in *Analytical Depth* and *Presentation Quality*, proving the effectiveness of our specialized agentic workflow.
> > > 2.  **Validation of Automated Metrics:** The strong positive correlation between human and LLM scoring (Pearson $r > 0.75$ for Total Score) **validates the reliability of the automated evaluation framework** used in our main paper. This confirms that our LLM-based judges serve as effective proxies for human experts in assessing complex financial reports.
> > > 3.  **Robust Inter-Rater Agreement:** The overall inter-rater reliability ($\alpha=0.64$) indicates a solid consensus among experts. Notably, the exceptionally high agreement in the *Presentation* dimension confirms that FinSight's multimodal capabilities (text-chart integration) provide a **clear, objectively recognizable advantage**. While agreement is slightly lower for Factual/Analytical dimensions due to the inherent subjectivity in evaluating financial analysis, the scores remain within the moderate agreement range ($\alpha > 0.4$), ensuring the validity of the results.
> > >
> > >
> > >
> > >
> > > #### 2. Report Accuracy (Golden Fact Recall)
> > > Directly measuring the factuality of long reports is challenging. To quantify this, we introduced a **Golden Facts Evaluation**. We extracted **13 core financial indicators** (Ground Truth) from the professional Golden Reports, covering Profitability (e.g., Gross Margin), Growth (e.g., Revenue Growth), Financial Health (e.g., Cash Flow), Valuation (e.g., PE Ratio), and Efficiency (e.g., ROE). Then, we manually verified how many of these specific data points were accurately retrieved and reported by each model across company-level tasks.
> > >
> > > **Table C: Key Information Recall**
> > >
> > > | Method | Avg. Hits (out of 13) | Avg. Recall Rate | Relative Performance |
> > > | :--- | :---: | :---: | :--- |
> > > | **FinSight (Ours)** | **7.1** | **54.6%** | -- |
> > > | Gemini-2.5-Pro Deep Research | 5.0 | 38.5% | -29.6% |
> > > | OpenAI Deep Research | 3.9 | 30.0% | -45.1% |
> > >
> > > Result shows that FinSight achieves a significantly higher recall rate, demonstrating superior coverage of critical financial data compared to commercial deep research systems.

---

> > > > ### Author Response · Authors · 2025-11-21
> > > >
> > > > #### 3. Citation Accuracy
> > > > We also manually verified the **Citation Faithfulness**. Experts checked the top 50 citations in generated reports to verify if the cited source actually supported the generated claim.
> > > >
> > > > **Table D: Citation Verification**
> > > >
> > > > | Metric | FinSight (Ours) | Gemini-2.5-Pro Deep Research |
> > > > | :--- | :---: | :---: |
> > > > | **Total Citations Checked** | 469 | 414 |
> > > > | **Overall Accuracy** | **72.92%** (342/469) | 69.81% (289/414) |
> > > >
> > > > Even while generating a higher volume of citations, FinSight maintains higher accuracy. We attribute this to our **Two-Stage Writing Framework** and generative retrieval mechanism, which identifies references during the drafting process rather than via post-hoc appending.
> > > >
> > > > ### Response to Q3: Definition of Financial Research Report
> > > >
> > > > Our definition of a "financial research report" targets **professional investment research** (similar to sell-side equity research), rather than general analytical summaries. This distinction is crucial and necessitates our complex multi-agent design. Specifically, these reports require:
> > > > 1.  **Multimodal Integration:** Seamless synthesis of text, professional charts, and tables (addressed by our Vision-Enhanced Mechanism).
> > > > 2.  **Analytical Depth:** Comprehensive coverage of a target's history, financials, and strategy, not just a brief summary (addressed by our Two-Stage Writing).
> > > > 3.  **Multi-source Heterogeneity:** Integration of real-time stock data, financial APIs, and news (addressed by our Variable Memory/CAVM).
> > > >
> > > > The Golden Reference reports used in our evaluation are actual deep research reports from professional securities firms, also confirming that our target is high-level professional analysis.
> > > >
> > > >
> > > > We once again thank you for your thorough and valuable suggestions. If you have any questions, please feel free to raise them, and we will do our best to address them.

---

> ### Author Response · Authors · 2025-11-26
> **Revision Update**
>
> Dear Reviewer:
>
> We would like to express our gratitude for your time and constructive feedback.
>
> Following your suggestions, we have just uploaded a revised version of our manuscript. To facilitate your review, we have **highlighted all major modifications in red**.
> In response to the constructive feedback received, we have made extensive revisions to both the main text and the appendix. We hope these substantial updates effectively address your concerns. We would greatly appreciate it if you could share your thoughts on the revised version. We are happy to engage in further discussions.
>
> Thank you for your time and effort in reviewing.
>
> Best regards, The Authors

---

### Author Response · Authors · 2025-11-25
**General Response**

We sincerely thank all the reviewers for their time and constructive feedback. We are thrilled that our core contributions have been recognized. Our FinSight approach has been highlighted as timely and practically relevant (Reviewer 2oEk), with a coherent system design and well-motivated integration of modalities (Reviewer 9Dq6), and a clear framework with effective two-stage writing strategy (vZ9e). We are particularly pleased that reviewers recognized the novelty of our CAVM architecture and Iterative Vision-Enhanced Mechanism (Reviewer 9Dq6, vZ9e), as well as the substantial engineering effort behind the system (Reviewer 2oEk).

The reviewers' primary concerns focused on several key areas, which we have addressed with detailed responses and extensive new quantitative analyses. We summarize these points below.

| Focus Area | Reviewer Concerns | Our Actions |
|:---|:---|:---|
| **1.Problem Formulation** | 2oEk | **Action: (1) Revised definition**: Updated the formulation of report R from a set to an ordered sequence of sections with hierarchical structure. **(2) Updated the formulation of generation process**: Provided explicit mathematical formulation of the mapping from query q to report R. |
| 2. Code Release & Reproducibility | 2oEk, vZ9e | **Action: Released complete source code**: The full implementation including multi-agent design and CAVM architecture is now available at https://anonymous.4open.science/r/FinSight-6739/. |
| 3. Human Evaluation & Validation | 2oEk, 9Dq6, vZ9e | **Action: (1) Comprehensive human expert study**: Recruited 6 graduate students with financial backgrounds to evaluate reports. FinSight achieves highest Total Score (7.11) vs. Gemini-2.5-Pro DR (6.11) and OpenAI DR (5.50). **(2) Human-LLM alignment**: Strong correlation (Pearson r=0.7587) validates automated metrics. **(3) Inter-rater reliability**: Krippendorff's α=0.6474 indicates solid expert consensus. |
| 4. LLM-as-Judge Reliability & Bias | 9Dq6, vZ9e | **Action: (1) Stability analysis**: Conducted 3 repeated runs with 95% CIs, showing low variance (std dev <1.0 across metrics). **(2) Cross-model verification**: Used GPT-5 as alternative evaluator; ranking consistency confirmed (Kendall's τ=0.764). **(3) Length-controlled evaluation**: FinSight (10k words) still outperforms baselines (6.93 vs. 6.82 vs. 6.11), proving gains are from quality not verbosity. |
| 5. Quantitative Metrics & Factual Accuracy | 9Dq6, vZ9e | **Action: (1) Golden Facts Evaluation**: FinSight achieves 54.6% recall rate on 13 core financial indicators vs. 38.5% (Gemini) and 30.0% (OpenAI). **(2) Citation verification**: Manual check of 469 citations shows 72.92% accuracy vs. 69.81% (Gemini). **(3) Visual quality metrics**: Implemented IQA metrics (Colorfulness, RMS Contrast, Edge Density) showing Iterative Vision Mechanism doubles scores across all dimensions. |
| 6. Novelty & CAVM Architecture | 2oEk, vZ9e | **Action: (1) Clarified distinctions**: Emphasized FinSight addresses data-intensive, multimodal long-report generation, not just domain extension. **(2) Detailed CAVM explanation**: Explained Variable Memory as novel state representation mechanism enabling executable representation and unified manipulation of data, tools, and agents. **(3) Code execution integration**: Provided comprehensive workflow showing how each agent uses code for data processing, dynamic retrieval, and context management. |
| 7. Benchmark Size & Generalization | 9Dq6 | **Action: (1) Justified benchmark construction**: Explained 20-sample benchmark covers diverse targets (companies/industries), 10 sectors, and 2 markets, ensuring high difficulty. **(2) Format consistency rationale**: Clarified that standardized formatting aligns with real-world professional standards. **(3) Future generalization plans**: Outlined domain-agnostic framework design enabling extension to other domains. |
| 8. Search Region Settings & Fairness | 9Dq6 | **Action: Clarified experimental setup**: All baselines evaluated via official interfaces with no control over internal settings. FinSight's China region setting aligns with benchmark (A-share/HK-stock). Fair comparison ensured via English prompts, with any bias likely favoring commercial baselines due to proprietary search resources. |

We believe these clarifications and additional quantitative results have thoroughly addressed the reviewers' concerns. We have provided comprehensive human evaluation, rigorous validation of automated metrics, detailed quantitative analyses, and full code release to ensure reproducibility and transparency.

Sincerely, our team has invested significant effort into this work. We truly hope that our response could encourage a more favorable reassessment of FinSight's significance, effectiveness, and completeness.

Best regards

---

### Note · Authors · 2025-12-01

I have read and agree with the venue's withdrawal policy on behalf of myself and my co-authors.